# Info-GRPO: Training Reasoning Models via Correlation-Aware Exploration

## Abstract

Recent studies have revealed policy collapse in advanced reasoning models trained with Group Relative Policy Optimization (GRPO), and entropy regularization has stood out as an elegant approach to promote exploration. Yet, within the vast token space of language models, entropy gradients often exhibit severe singularities, creating a direct conflict with the natural entropy decay required for convergence and thereby disturbing optimization dynamics. To resolve this tension, we present Info-GRPO, an information-theoretic framework that reconciles the opposing entropic forces of exploration and convergence by cultivating correlation between the policy and a latent prior. Info-GRPO leverages a contrastive regularization that maximizes the mutual information between latent variables and the policy. Intuitively, by augmenting prompts with latent variables, the model explores a more diverse set of policies that remain correlated with the latent prior, guiding conditional entropy toward convergence. Through this correlation-aware design, Info-GRPO respects the natural entropy reduction during training while enabling more effective exploration. Extensive experiments demonstrate that Info-GRPO significantly outperforms vanilla GRPO and entropy-regularized GRPO across diverse reasoning benchmarks. For instance, it achieves improvements of 3.75%, 1.66%, and 4.16% in Avg@8 compared to GRPO based on Qwen2.5-Math-7B, Qwen2.5-7B, and DeepSeek-R1-Distill-Qwen-7B, respectively, under the AIME24 benchmark. Furthermore, analysis reveals that Info-GRPO induces distinct and interpretable reasoning patterns conditioned on the latent variable, showcasing a more systematic and effective exploration strategy.

## 1 Introduction

Recent advances in large language models (LLMs) (OpenAI, 2024; 2025; Anthropic, 2025; Team, 2025a; Guo et al., 2025) have ushered in a new era of sophisticated reasoning capabilities, driving performance to new heights across a variety of complex domains such as mathematics and programming (Team, 2025b; Yang et al., 2025). These models increasingly rely on advanced reinforcement learning paradigms to refine their reasoning processes and align them with desired outcomes. A key driver behind this progress is Reinforcement Learning with Verifiable Rewards (RLVR) (Lambert et al., 2024), a paradigm that scalably rewards outcomes against ground-truth solutions by leveraging external verification signals, effectively bypassing the need for labor-intensive supervision. Building on this, methods like Group Relative Policy Optimization (GRPO) (Shao et al., 2024) have further enhanced the stability and sample efficiency of RLVR by introducing group-relative advantage estimation, which accelerates convergence without significant computational overhead.

Despite the empirical success of RLVR and GRPO, these methods remain fundamentally limited in their ability to encourage exploration beyond the model's pre-existing knowledge, which severely limits the potential for improvement under diverse sampling conditions (Ma et al., 2025). Inherently constrained by their on-policy nature, these approaches predominantly reinforce reasoning paths that the model already deems highly rewarding. As the model grows increasingly confident in its predictions, exploration is progressively reduced (Walder & Karkhanis, 2025). This becomes particularly acute in environments with sparse rewards or deceptive local optima, where the model is highly susceptible to converging prematurely toward suboptimal solutions (Hong et al., 2018). With training progression, models optimized with RLVR often exhibit policy collapse (He et al., 2025), becoming overconfident in a narrow set of strategies and sacrificing policy diversity.

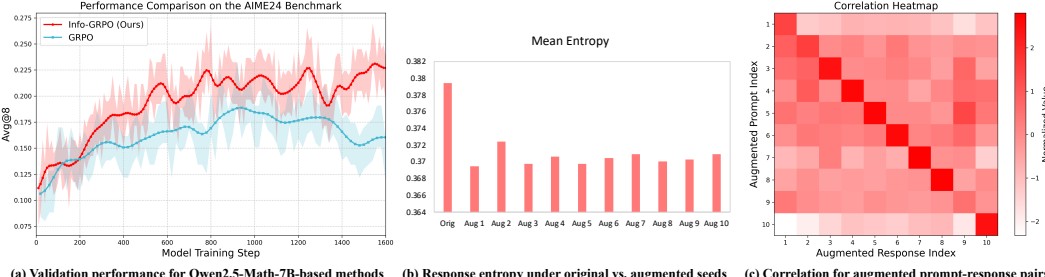

(a) Validation performance for Qwen2.5-Math-7B-based methods    (b) Response entropy under original vs. augmented seeds    (c) Correlation for augmented prompt-response pairs

Figure 1: (a) Validation results on Qwen2.5-Math-7B show that Info-GRPO significantly outperforms the GRPO baseline. (b) Info-GRPO encourages mild high-entropy policies from original prompts and low-entropy policies from seed-augmented prompts. (c) The high correlation between random seeds and their respective outputs demonstrates a diversity-driven exploration mechanism.

Entropy regularization has emerged as a remedy to mitigate policy collapse and encourage diversity in recent reasoning models (Yu et al., 2025; Cui et al., 2025; Wang et al., 2025a). Rooted in conventional RL, entropy helps balance exploration and exploitation by preventing early convergence to deterministic policies (Ziebart et al., 2008; O'Donoghue et al., 2016; Haarnoja et al., 2018). Common strategies include adding entropy bonuses to rewards or advantages (Chen et al., 2025; Cheng et al., 2025; Wang et al., 2025b) or targeting high-entropy regions during sampling to improve coverage of uncertain decisions (Wang et al., 2025a; Zheng et al., 2025b). These approaches help the model explore uncertain regions of the policy space and avoid overconfidence in suboptimal strategies.

Nevertheless, incorporating entropy regularization into GRPO introduces a fundamental tension with the natural trajectory of optimization. *The primary training objective drives entropy reduction as a prerequisite for convergence, compelling the model to learn a confident, high-quality policy (Fu et al., 2025). In direct opposition, the regularization term actively pulls the policy toward higher entropy for the sake of exploration.* This creates an unstable dynamic where the optimization oscillates between two conflicting objectives. *The instability is severely exacerbated in the vast token spaces of modern LLMs (e.g., about 152,000 tokens (Yang et al., 2025)), where the gradients from the entropy term are highly susceptible to singularities* (as elaborated in Sec. 4). Although the trade-off between opposing forces can be managed via careful hyperparameter tuning, such adjustments are merely a heuristic compromise. This highlights the need for a new framework that fosters diversity without relying on this inherently unstable mechanism.

In this paper, we propose Info-GRPO, a novel training framework that reframes the challenge of exploration in policy optimization. Rather than balancing exploration and convergence as opposing objectives in entropy regularization, we introduce a correlation-aware perspective inspired by information theory. Info-GRPO addresses the inherent tension between entropy-driven exploration and entropy-reducing convergence by conditioning the policy on a latent prior and explicitly optimizing the statistical dependency between them. This is achieved by maximizing the mutual information between the latent variable and the policy, which simultaneously encourages a diverse set of strategies that are each highly confident, as shown in Fig. 1. In summary, our contributions are as follows:

- We show that naive entropy regularization in large vocabularies suffers from gradient singularities induced by massive tokens in the tail of the distribution. By framing entropy regularization as a special case of mutual information, we reorient learning toward conditional entropy reduction, yielding gradients inherently consistent with convergence.

- We introduce Info-GRPO, a correlation-aware training paradigm that augments prompts with latent variables and employs a mutual information objective to correlate distinct reasoning strategies with latent priors. This simple and effective design resolves the tension between exploration and convergence with a coherent trajectory of entropy reduction.

- We conduct evaluations across diverse benchmarks and models, demonstrating that Info-GRPO consistently and substantially outperforms GRPO baselines. Furthermore, latent-conditioned outputs exhibit distinct and interpretable reasoning patterns, providing direct evidence of structured exploration and a novel pathway for steering and analyzing cognitive strategies in large models.

## 2 RELATED WORK[1]

**Reinforcement Learning for LLM Reasoning.** Reinforcement learning has become pivotal for improving LLM reasoning (OpenAI, 2024; 2025; XAI, 2024; Qwen, 2024; Guo et al., 2025). Methods like RL from Human Feedback (RLHF) Ouyang et al. (2022) and those based on Verifiable Rewards (RLVR) Lambert et al. (2024); Shao et al. (2025) are foundational. A significant advancement is the value-model-free Group Relative Policy Optimization (GRPO) Shao et al. (2024), which enhances stability and efficiency compared to PPO Schulman et al. (2017b). Recent works, such as Dr. GRPO Liu et al. (2025), refine the advantage estimation, and GSPO Zheng et al. (2025a) extends GRPO to stabilize Mixture-of-Experts training. Despite these successes, GRPO-based approaches fundamentally suffer from insufficient exploration and policy collapse Chen et al. (2025). Our work focuses on extending GRPO to achieve improvements in exploration and exploitation capabilities.

**Entropy Regularization for Reinforcement Learning.** Entropy regularization is an essential technique in RL to address premature convergence and insufficient exploration (Hong et al., 2018; Cui et al., 2025). In LLM fine-tuning, recent works have utilized entropy to drive reasoning behavior Wang et al. (2025a), build intermediate feedback via targeted rollouts Zheng et al. (2025b), augment the advantage function for diversity-driven exploration (Chen et al., 2025; Cheng et al., 2025), or mitigate premature convergence through appropriate control He et al. (2025); Wang et al. (2025b). While effective, it remains challenging for entropy-based methods since the uncertainty introduced to promote exploration may weaken confidence. In this paper, we augment entropy-regularized GRPO with mutual information to address the contradiction between diversity and confidence.

**Mutual Information for Structured Exploration.** Mutual Information (MI) has been widely used to capture dependencies between random variables and to introduce structure into learned representations Hjelm et al. (2018); Poole et al. (2019); Wen et al. (2020); Rakelly et al. (2021); Chen et al. (2024). It plays a central role in representation disentanglement Chen et al. (2016); Zhao et al. (2017) and in self-supervised contrastive learning via the InfoNCE objective Chen et al. (2020); Zhang et al. (2023); Lee et al. (2024). In the context of LLMs, MI is increasingly leveraged for reasoning control: it has been used to downweight redundant completions in GRPO-based methods Chen et al. (2025), to analyze reasoning trajectory dynamics Qian et al. (2025), and to optimize conditional MI for preference alignment Xiao et al. (2025). Such controllability enables more structured and targeted exploration rather than purely stochastic diversity. Motivated by these insights, we explore the integration of MI maximization with RL techniques. By maximizing the mutual information conditioned on a latent variable, our method induces structured exploration through latent–policy correlation, maintaining policy diversity while ensuring training stability.

## 3 BACKGROUND AND NOTATIONS

In RLVF, we model the LLM as a policy $\pi_{\boldsymbol{\theta}}$. The generation process begins with an initial state $s_0$, which corresponds to the input prompt. At each subsequent step $t$, the state is defined by the history of previous actions, $s_t = (s_0, a_0, \ldots, a_{t-1})$, based on which the policy $\pi_{\boldsymbol{\theta}}(\cdot|s_t)$ selects an action $a_t$ (a token). The full sequence of actions $\tau = (a_0, a_1, \ldots)$ constitutes the complete trajectory.

**Proximal Policy Optimization (PPO)** is a foundational on-policy algorithm for LLM fine-tuning, prized for its stability and reliability (Schulman et al., 2017b). It addresses the sensitivity to step size inherent in traditional policy gradient methods. PPO stabilizes training by optimizing a clipped surrogate objective that depends on an advantage estimate. Critically, standard PPO requires a separate, trainable critic model to compute this advantage, which can be computationally expensive.

**Group Relative Policy Optimization (GRPO)** is an efficient, critic-free alternative (Shao et al., 2024). Instead of training a critic, GRPO estimates the advantage $\hat{A}$ for an entire trajectory $\tau$ by comparing its reward to that of other trajectories in a sampled group $\mathcal{T} = \{\tau^i\}_{i=1}^{G}$. This critic-free advantage is incorporated into a PPO-style clipped objective. For a trajectory $\tau$, the objective is:

$$J_{\text{GRPO}}(\boldsymbol{\theta}, \mathcal{T}) = \sum_{\tau \in \mathcal{T}} \sum_t \min\left(r_t(\boldsymbol{\theta})\hat{A}(\tau), \text{clip}(r_t(\boldsymbol{\theta}), 1 - \epsilon, 1 + \epsilon)\hat{A}(\tau)\right), \quad (1)$$

---

[1]More discussion of related work is provided in Appendix A.2.

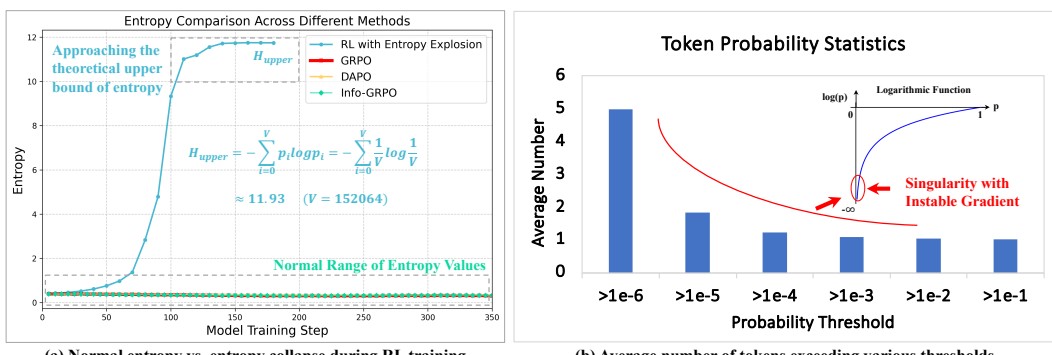

(a) Normal entropy vs. entropy collapse during RL training      (b) Average number of tokens exceeding various thresholds

Figure 2: (a) An entropy bonus can degrade the policy into a uniform distribution, where entropy reaches its upper bound. (b) This instability is inherent to LLMs, whose vocabularies are dominated by a vast tail of low-probability tokens that induce logarithmic singularities (see Proposition 1). Statistically, in the 152,064 tokens of Qwen2.5-7B, fewer than five tokens exceed a probability of $10^{-6}$. The instability makes direct entropy maximization an ill-posed objective for exploration.

where $r_t(\boldsymbol{\theta}) = \frac{\pi_{\boldsymbol{\theta}}(a_t|s_t)}{\pi_{\boldsymbol{\theta}_{\text{old}}}(a_t|s_t)}$ is the probability ratio for taking action $a_t$ in state $s_t$. This approach significantly simplifies the training process by removing the need for a separate critic model.

**Policy Entropy** is a fundamental concept that measures uncertainty of policy in RL. For a initial state $s_0$, the Shannon entropy of a policy trajectory $\tau$ is defined as:

$$H(\pi_{\boldsymbol{\theta}}(\mathcal{T}|s_0)) := \sum_{\tau \in \mathcal{T}} \sum_{t} H(\pi_{\boldsymbol{\theta}}(\cdot|s_t)), \ H(\pi_{\boldsymbol{\theta}}(\cdot|s_t)) = -\sum_{a \in V} \pi_{\boldsymbol{\theta}}(a|s_t) \log \pi_{\boldsymbol{\theta}}(a|s_t), \quad (2)$$

where $V$ is the set of all possible actions, *i.e.*, the vocabulary in the context of LLMs, and $\pi_{\boldsymbol{\theta}}(a|s)$ is the probability of selecting action $a$ in state $s$. A high entropy value corresponds to a more uniform, uncertain policy that encourages exploration, while a low entropy value signifies a more deterministic, confident policy geared towards exploitation.

## 4 WHY ENTROPY REGULARIZATION IS INTRACTABLE IN LLMS

### 4.1 POLICY ENTROPY IN REINFORCEMENT LEARNING

During the fine-tuning of large reasoning models, a monotonic decrease in policy entropy is an expected outcome of successful learning. The seminal work of (Cui et al., 2025) provides a deep analysis of this phenomenon. They establish a strong positive correlation between the probability of an action under the policy and its corresponding advantage value. As the model learns to identify high-quality reasoning paths (Fu et al., 2025), it naturally assigns higher probabilities to actions with high advantages, and vice versa.

**Entropy Collapse and Regularization.** While entropy reduction signifies learning, a pathological version of this process, known as entropy collapse, refers to a sharp drop in policy entropy at the very beginning of training (Hong et al., 2018; He et al., 2025; Cheng et al., 2025). Such a rapid decrease leads to premature convergence, where the model becomes overconfident in a suboptimal strategy and insufficient exploration of the vast solution space. To counteract this problem, entropy regularization has become an essential technique in modern RL (Hong et al., 2018; He et al., 2025) to maintain sufficient policy diversity to prevent premature convergence. This is typically implemented as a token-level entropy bonus added to the primary objective (*i.e.*, maximizing Eq. (2)), ensuring the model retains its exploratory capacity throughout the fine-tuning process.

## 4.2 SINGULARITY TRAP OF ENTROPY REGULARIZATION IN THE CURSE OF SCALE

The RL objective seeks certainty by reducing entropy, while the entropy bonus pursues possibility by increasing it, creating a fundamental conflict that destabilizes optimization. As demonstrated by (Cui et al., 2025; He et al., 2025), managing this tension with a simple coefficient is fraught with difficulty: **small coefficients have a negligible effect on exploration, while large ones risk catastrophic instability and entropy explosion**. This is also evident in our Fig. 2(a), where we use a relative small coefficient of $0.05$, which suggests that applying entropy regularization to large-scale models is a non-trivial challenge that goes beyond simple hyperparameter tuning.

We demonstrate that this tension manifests as a concrete and severe numerical instability caused by the logarithmic singularity $\log \pi_{\boldsymbol{\theta}}(y|s)$, and this is not an incidental artifact but an essential flaw rooted in the high-dimensional, sparse nature of LLM vocabularies.

**Proposition 1** (The Singularity Trap for High-Dimensional Entropy Maximization). *Let $\pi_{\boldsymbol{\theta}}(\cdot|s)$ be a policy over a discrete vocabulary $A$ of size $V$, and let the policy entropy gradient be $\nabla_{\theta} H(\pi_{\boldsymbol{\theta}}) = -\sum_{a \in A} \nabla_{\theta} \pi_{\boldsymbol{\theta}}(a|s)(1 + \log \pi_{\boldsymbol{\theta}}(a|s))$. The gradient is fundamentally ill-conditioned in high-dimensional spaces, characterized by two results[2]:*

*(1) **Quantitative Bound of the Tail Set:** For any probability threshold $\delta \in (0, 1)$, the set of low-probability "tail" tokens, $A_{\delta} := \{a \in A \mid \pi_{\boldsymbol{\theta}}(a|s) < \delta\}$, constitutes the vast majority of the vocabulary. Its size is lower-bounded by:*

$$|A_{\delta}| \geq V - \frac{1}{\delta} \tag{3}$$

*(2) **Quantitative Bound on Gradient Instability:** Consequently, the gradient contribution from this tail, $\nabla_{\theta} H(\pi_{\boldsymbol{\theta}})_{tail} = -\sum_{a \in A_{\delta}} \nabla_{\theta} \pi_{\boldsymbol{\theta}}(a|s)(1 + \log \pi_{\boldsymbol{\theta}}(a|s))$, is numerically unstable. The cumulative magnitude of its logarithmic scaling factors, defined as the Total Tail Instability (TTI), has a lower bound that grows linearly with $V$:*

$$TTI := \sum_{a \in A_{\delta}} |1 + \log \pi_{\boldsymbol{\theta}}(a|s)| \geq \left(V - \frac{1}{\delta}\right)|1 + \log \delta| \tag{4}$$

*These results hold provided $V > 1/\delta$, a condition readily met in LLMs, confirming that the entropy gradient is structurally unstable.*

**Remark 1** *A large $V$ makes a dominant tail set inevitable*. Claim 1 formalizes that a massive vocabulary must result in an extremely sparse distribution, and the number of tail tokens grows linearly with $V$. In the context of LLMs, this means the region where the problematic $\log \pi_{\boldsymbol{\theta}}(a|s)$ term can cause instability is not a fringe case but constitutes nearly the entire action space. For a concrete example, Fig. 2(b) illustrates this phenomenon using the sampling distribution of Qwen2.5-7B, confirming the prevalence of a dominant tail set.

**Remark 2** *A dominant tail set makes gradient anomalies inevitable*. Claim 2 shows that the cumulative explosive potential from the tail tokens also grows linearly with $V$. The final gradient vector becomes an aggregation of tens of thousands of ill-conditioned terms, where the learning signal from the few important "head" tokens is inevitably drowned out by the numerical noise from the vast tail.

**Remark 3** *Entropy gradient is **asymmetrically** unstable*. During entropy maximization, the singularity creates a powerful amplifying force that increases countless near-zero probabilities, leading to explosive updates and uniform distribution. Conversely, for entropy minimization, the singularity creates a suppressive force that pushes these negligible values closer to their lower bound of zero.

In summary, the vast token space is the direct cause of the singularity trap, which transforms entropy regularization from a manageable technique into a barrier for LLMs. While encouraging exploration via an entropy bonus is precarious, driving an LLM's policy toward certainty is reliable. This motivates our search for an alternative exploration mechanism that avoids this intractable dynamic.

---

[2]See Theorem 1 and 2 in Appendix A.1

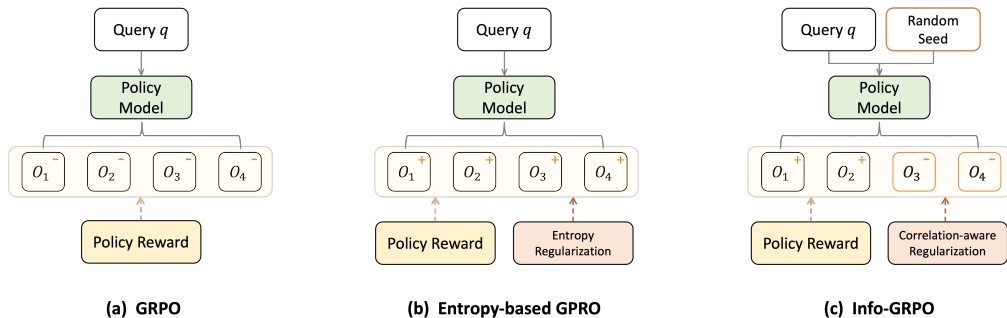

Figure 3: An intuitive comparison of exploration methods. Plus (+) and minus (-) signs represent forces that increase or decrease policy entropy. (a) Vanilla GRPO focuses on exploitation (-), often leading to policy collapse. (b) Entropy regularization promotes diversity by increasing all entropy (+), but this risks collapsing the model towards a uniform distribution. (c) Info-GRPO balances a mild entropy increase (+) with a strong, seed-conditioned entropy decrease (-). This maximizes the mutual information between seeds and outcomes, using correlation to drive stable exploration.

## 5 THE INFO-GRPO FRAMEWORK

This analysis of the singularity trap directly motivates our solution in a principled manner. Instead of pursuing the numerically unstable objective of maximizing marginal entropy, we propose its most direct information-theoretic extension: maximizing the mutual information (MI), $I(\tau, z) = H(\pi) - H(\pi|z)$. This objective effectively shifts the unstable maximization of marginal entropy to the stable minimization of conditional entropy. As established in Remark 3, this minimization dynamic is robust and does not suffer from the singularity trap, thereby resolving the core instability. Furthermore, this MI objective represents a minimal algorithmic modification that reuses the existing entropy computation module on a latent-augmented batch, allowing us to isolate the benefits of the proposed strategy from other confounding factors.

### 5.1 FROM ENTROPY REGULARIZER TO CORRELATION-AWARE REGULARIZER

To resolve the tension between exploration and convergence, we introduce an improved regularizer that conditions the policy on a latent variable $z \sim p(Z)$. As shown in Fig. 3, our objective balances two competing entropic forces for stable and structured exploration:

$$\max_{\boldsymbol{\theta}} \ \mathcal{J}_{\text{Info}}(\boldsymbol{\theta}) = \max_{\boldsymbol{\theta}} \ \left( \alpha \cdot H(\pi_{\boldsymbol{\theta}}(\mathcal{T} \mid s_0)) - H(\pi_{\boldsymbol{\theta}}(\mathcal{T} \mid s_0, z)) \right). \tag{5}$$

The objective is driven by a consolidation term, $-H(\pi_{\boldsymbol{\theta}}(\tau \mid s_0, z))$, which compels each latent-conditioned policy to converge to a confident strategy. This is counterbalanced by the weighted entropy bonus, in which the coefficient $\alpha \in [0, 1)$ mitigates gradient instability, and a small positive value is retained to anchor the exploration in the model's original policy.

**Correlation with Mutual Information.** Our objective in Eq. (5) is deeply grounded in information theory. For the case of $\alpha = 1$, it becomes equivalent to maximizing the mutual information (MI) between the trajectory $\tau$ and the latent variable $z$:

$$I(\tau, z \mid s_0) = \mathbb{E}_{\tau \sim \pi_{\boldsymbol{\theta}}(\cdot \mid s_0, z), z \sim p(Z)} \left[ \log \frac{\pi_{\boldsymbol{\theta}}(\tau, z \mid s_0)}{p(z) \, \pi_{\boldsymbol{\theta}}(\tau \mid s_0)} \right]. \tag{6}$$

This perspective reframes exploration: rather than pursuing a single, high-entropy policy, we maximize the statistical **correlation** between latent codes and their corresponding reasoning paths. Our framework thus cultivates a diverse ensemble of low-entropy policies, each activated by a different $z$. Using correlation to orchestrate diversity provides a stable and principled mechanism for exploration, resolving the exploration and exploitation dilemma.

---

**Algorithm 1** The Info-GRPO Training Algorithm

---

1: **Input:** Initial policy $\pi_{\boldsymbol{\theta}}$, dataset $\mathcal{D}$, hyperparameters $G, \lambda, \alpha$.
2: **Initialize:** Policy parameters $\boldsymbol{\theta}$.
3: **for** each training iteration **do**
4:     Sample initial states $\{s_0^{(i)}\} \subset \mathcal{D}$ and latents $\{z^{(i)}\} \sim p(Z)$.
5:     Sample trajectories: $\mathcal{T}_{\text{ori}} = \{\pi_{\boldsymbol{\theta}}(\cdot \mid s_0^{(i)})\}$ and $\mathcal{T}_{\text{aug}} = \{\pi_{\boldsymbol{\theta}}(\cdot \mid s_0^{(i)}, z^{(i)})\}$.
6:     Calculate the objective $J_{\text{Info-GRPO}}(\boldsymbol{\theta})$ per Eq. (7).
7:     Update $\boldsymbol{\theta}$ using gradient ascent: $\boldsymbol{\theta} \leftarrow \boldsymbol{\theta} + \eta \nabla_{\boldsymbol{\theta}} J_{\text{Info-GRPO}}$.
8: **end for**
9: **Return:** Optimized parameters $\boldsymbol{\theta}$.

---

## 5.2 IMPLEMENTATION

The Info-GRPO framework modifies the standard GRPO (Shao et al., 2024) training loop by introducing a latent-augmented sampling strategy and a corresponding correlation-aware objective function. The implementation is designed to be efficient and minimally invasive.

**Latent-Augmented Sampling.** Unlike vanilla GRPO, which samples a single group of trajectories, Info-GRPO generates two distinct groups for each initial state $s_0$. First, a group of original trajectories, $\mathcal{T}_{\text{ori}}$, is sampled from the base policy $\pi_{\boldsymbol{\theta}}(\cdot \mid s_0)$. Concurrently, the policy is conditioned on a latent variable $z$, which is a discrete token sampled uniformly from a predefined set (*e.g.*, $\{1, 2, \ldots, 10\}$). The conditioning is achieved by integrating $z$ into $s_0$ using a deterministic textual template, such as appending the string: *"Choosing random seed $\{z\}$ from seed list 1 to 10."* A second group of augmented trajectories, $\mathcal{T}_{\text{aug}}$, is then sampled from this conditioned policy, $\pi_{\boldsymbol{\theta}}(\cdot \mid s_0, z)$.

**The Info-GRPO Objective.** The complete objective function combines the GRPO policy loss, computed over the unified set of trajectories, with the correlation-aware regularizer. The regularizer's entropy terms are estimated via Monte Carlo approximation using their respective trajectory sources $\mathcal{T}_{\text{ori}}$ for the marginal entropy and $\mathcal{T}_{\text{aug}}$ for the conditional entropy. The final objective to be maximized is:

$$J_{\text{Info-GRPO}}(\boldsymbol{\theta}) = J_{\text{GRPO}}(\boldsymbol{\theta}, \mathcal{T}_{\text{ori}} \cup \mathcal{T}_{\text{aug}}) + \lambda \left( \alpha \cdot H(\pi_{\boldsymbol{\theta}}(\mathcal{T}_{\text{ori}} \mid s_0)) - H(\pi_{\boldsymbol{\theta}}(\mathcal{T}_{\text{aug}} \mid s_0, z)) \right). \quad (7)$$

Here, $J_{\text{GRPO}}$ is the standard clipped objective defined in Eq. (1), with its advantage estimates $\hat{A}(\tau)$ computed across the entire merged set $\mathcal{T}_{\text{ori}} \cup \mathcal{T}_{\text{aug}}$ for robust estimation. The $\lambda$-weighted term is the practical implementation of our regularizer, directly guiding the model to cultivate a diverse yet confident policy space.

## 6 EXPERIMENTS

### 6.1 TRAINING DETAILS

We conduct experiments on three open-source models: Qwen2.5-7B (Team, 2024), Qwen2.5-Math-7B (Yang et al., 2024), and DeepSeek-R1-Distill-Qwen-7B (Guo et al., 2025). (1) For Qwen2.5 series models, the RL training set and prompts follow DAPO-Math-17K (Yu et al., 2025), which contains 17,917 questions, each paired with an integer as its corresponding answer. The max token length is 4,096, following the official model configuration of Qwen2.5-Math-7B. (2) For DeepSeek-R1-Distill-Qwen-7B, the RL training set and prompts are sourced from (He et al., 2025), with 48,371 samples. The max token length is 8,192. For both training settings, the learning rate is 1e-6 with a batch size of 128. We set $\lambda = 0.005$ and $\alpha = 0.5$ throughout to mitigate the potential for gradient instability from the entropy maximization term. In each rollout step, 16 responses are sampled per prompt with a temperature of 1.0. Regarding the two SOTA methods, the target entropy is 0.2 for Skywork-OR1 (He et al., 2025). The clipping parameter $\epsilon_{low} = 0.2$ and $\epsilon_{high} = 0.28$ for DAPO (Yu et al., 2025). The models are trained on 8 NVIDIA B200 GPUs, and the best results are reported. More model series (*e.g.*, DeepSeek-R1-Distill-Llama-8B (Guo et al., 2025)), domains (*e.g.*, Code), and comparative experiments with more SOTA methods are provided in the Appendix A.3.

Table 1: Comparison of methods on different backbones and benchmarks ($Avg@8$, %).

| Backbone | Method | AIME24 | AIME25 | AMC | MATH500 | Minerva | **Average** |
|---|---|---|---|---|---|---|---|
| *Qwen2.5-7B* | Base Model | 12.08 | 7.50 | 42.77 | 75.50 | 35.06 | 34.58 |
| | GRPO | 20.42 | 10.83 | 56.48 | **80.53** | 37.82 | 41.22 |
| | DAPO | 18.75 | **11.25** | 48.95 | 78.13 | 36.53 | 38.72 |
| | Skywork-OR1 | 20.83 | 7.50 | 57.38 | 78.38 | 38.05 | 40.43 |
| | Info-GRPO (Ours) | **22.08** | 8.75 | **58.28** | 79.60 | **38.37** | **41.42** |
| *Qwen2.5-Math-7B* | Base Model | 11.67 | 8.33 | 48.64 | 82.38 | 36.40 | 37.48 |
| | GRPO | 21.25 | 11.67 | 66.57 | **86.75** | 38.97 | 45.04 |
| | DAPO | 21.25 | 7.92 | 79.52 | 86.35 | 38.92 | 46.79 |
| | Skywork-OR1 | 22.50 | 13.33 | **81.02** | 84.73 | 38.19 | 47.95 |
| | Info-GRPO (Ours) | **25.00** | **15.83** | 78.46 | 86.63 | **39.34** | **49.05** |
| *DeepSeek-R1-Distill-Qwen-7B* | Base Model | 54.58 | 34.17 | 81.63 | 92.03 | 39.89 | 60.46 |
| | GRPO | 57.92 | 34.17 | 82.38 | 93.63 | 44.16 | 62.45 |
| | DAPO | 58.33 | 37.50 | 82.53 | 93.63 | 44.53 | 63.30 |
| | Skywork-OR1 | 59.58 | 37.92 | 82.08 | 93.73 | 44.12 | 63.48 |
| | Info-GRPO (Ours) | **62.08** | **45.83** | **83.13** | **93.98** | **44.94** | **65.99** |

## 6.2 BENCHMARKS AND METRICS

**Accuracy.** The methods are validated on AIME 2024, AIME 2025 (Li et al., 2024), AMC (Li et al., 2024), MATH500 (Hendrycks et al., 2021), and Minerva (Lewkowycz et al., 2022) benchmarks, with the test sets containing 30, 30, 83, 500, and 272 samples, respectively. During evaluation, the rollout temperature is 0.6. Following (He et al., 2025), $Pass@K$ is used to measure the reasoning ability of the model. For a given question, $Pass@K = 1$ if at least one of the $K$ sampled outputs passes verification, and 0 otherwise. For stability, each test sample is repeated eight times to compute $Pass@1$, $Pass@8$, and $Avg@8$, which is the average of $Pass@1$.

**Diversity.** To report the influence of the latent seed on generation, we quantify the coupling between a seed and its corresponding trajectory. We define the score as the average log-probability of the trajectory, given the initial state $s_0$ and the specific seed $z$:

$$\text{Correlation}(z, \tau \mid s_0) = \frac{1}{|\tau|} \sum_t \log \pi(a_t \mid s_0, z, a_{<t}). \tag{8}$$

## 6.3 COMPARATIVE RESULTS

**Comparisons on multiple benchmarks.** Table 1 shows that Info-GRPO achieves state-of-the-art performance, securing the top average score on each backbone: 41.42% (Qwen2.5-7B), 49.05% (Qwen2.5-Math-7B), and 65.99% (DeepSeek-R1-Distill-Qwen-7B). For example, Info-GRPO outperforms GRPO by an average of 4.01% and DAPO by 2.26% with Qwen2.5-math, indicating its robustness and general effectiveness across different pre-trained models. For demanding reasoning benchmarks, Info-GRPO underperforms on AIME25 based on Qwen2.5-7B, presumably owing to the limited capabilities of the base model, which constrain the entropy-regularized methods on this most challenging dataset. An in-depth analysis is provided in the Appendix A.4. This limitation is alleviated as the base model's capabilities improve, such as the best value of 15.83% and 45.83% on the other two backbones.

**Comparison of multiple metrics based on multiple backbones.** As shown in Table 2, we conducted a multi-metric evaluation on AIME24. Info-GRPO achieves the top $Pass@1$ score of 30.00% on Qwen2.5-Math-7B, but not good on DeepSeek-R1-Distill backbone. This may be because the benefits of a diversity-driven exploration strategy are less pronounced under the single-attempt constraint of this metric, particularly when the base model's capability is already strong. On $Pass@8$, Info-GRPO performs optimally except for the Qwen2.5-Math-based method, where it performs suboptimally. Crucially, when measured by $Avg@8$, which is a more robust metric, Info-GRPO outperforms all competing methods across all three backbones without exception. For instance, it outperforms GRPO by 3.75%, 1.66%, and 4.16% on Qwen2.5-Math-7B, Qwen2.5-7B, and DeepSeek-R1-Distill-Qwen-7B, respectively, further validating the method's consistent superiority.

Table 2: Comparison of methods based on different backbones on the AIME 2024 benchmark (%).

| Metric | Backbone | Base | GRPO | DAPO | Skywork-OR1 | **Info-GRPO** |
|--------|----------|------|------|------|-------------|---------------|
| | Qwen2.5-7B | 13.33 | 16.67 | 13.33 | 16.67 | **16.67** |
| *Pass@1* | Qwen2.5-Math-7B | 13.33 | 23.33 | 26.67 | 26.67 | **30.00** |
| | DeepSeek-R1-Distill-Qwen-7B | 53.33 | **66.67** | 63.33 | 63.33 | 56.67 |
| | Qwen2.5-7B | 30.00 | 30.00 | 33.33 | 33.33 | **36.67** |
| *Pass@8* | Qwen2.5-Math-7B | 23.33 | **43.33** | 30.00 | 33.33 | 36.67 |
| | DeepSeek-R1-Distill-Qwen-7B | 80.00 | 83.33 | 83.33 | 80.00 | **86.67** |
| | Qwen2.5-7B | 12.08 | 20.42 | 18.75 | 20.83 | **22.08** |
| *Avg@8* | Qwen2.5-Math-7B | 11.67 | 21.25 | 21.25 | 22.50 | **25.00** |
| | DeepSeek-R1-Distill-Qwen-7B | 54.58 | 57.92 | 58.33 | 59.58 | **62.08** |

Table 3: Ablation on coefficient $\lambda$. 'P1', 'P8', 'A8' are $Pass@1$, $Pass@8$, and $Avg@8$.

| Coef | AIME24 | | | AIME25 | | |
|------|--------|----|----|--------|----|----|
| | P1 | P8 | A8 | P1 | P8 | A8 |
| 0.5 | 50.00 | 80.00 | 57.08 | 36.67 | 63.33 | 39.58 |
| 0.05 | **56.67** | 83.33 | 58.75 | 30.00 | 60.00 | 40.42 |
| 0.005 | **56.67** | **86.67** | **62.08** | 50.00 | 63.33 | **45.83** |
| 0.002 | **56.67** | 83.33 | **62.08** | **53.33** | **66.67** | 44.17 |

Table 4: Ablation on max response lengths based on DeepSeek-R1-Distill-Qwen-7B.

| Len | AIME24 | | | AIME25 | | |
|-----|--------|----|----|--------|----|----|
| | P1 | P8 | A8 | P1 | P8 | A8 |
| 2K | 43.33 | 80.00 | 56.67 | 33.33 | 60.00 | 37.08 |
| 3K | 46.67 | 83.33 | 61.25 | 30.00 | 63.33 | 37.92 |
| 4K | 53.33 | 83.33 | 60.00 | 40.00 | **66.67** | 40.42 |
| 8K | **56.67** | **86.67** | **62.08** | **50.00** | 63.33 | **45.83** |

## 6.4 ABLATION STUDY

**Analysis of Different Coefficients.** Based on DeepSeek-R1-Distill-Qwen-7B, Table 3 summarizes the impact of the regularization coefficient $\lambda$ on model performance, revealing a clear advantage for smaller values. Lower coefficients like 0.005 and 0.002 consistently enhance generation quality and stability by reducing output stochasticity. This trend is reflected in the multi-sample metrics. For instance, on AIME24, a coefficient of 0.005 achieves the highest $Pass@8$ (86.67%) and $Avg@8$ (62.08%) scores. This pattern holds on AIME 2025, where the 0.002 coefficient yields the best $Pass@8$ result (66.67%). The strong $Pass@1$ performance further confirms that this constrained exploration also benefits single-sample reliability. Based on its robust results across benchmarks, we selected a coefficient of 0.005 as the optimal setting for our experiments.

**Analysis of different max response lengths.** Table 4 investigates the impact of the maximum response length on reasoning performance. The results indicate a clear positive correlation between longer response allowances and improved performance across both AIME benchmarks. Increasing the maximum length from 2K to 8K tokens leads to consistent gains. On AIME 2024, the 8K setting achieves the highest scores across all metrics: $Pass@1$ (56.67%), $Pass@8$ (86.67%), and $Avg@8$ (62.08%). A similar trend is observed on the more challenging AIME 2025, where the 8K length yields the best $Pass@1$ (50.00%) and $Avg@8$ (45.83%), while $Pass@8$ peaks at 66.67% with a 4K length. These findings demonstrate that a more generous response length is critical for complex reasoning tasks, as it provides the model with sufficient capacity to elaborate on logical steps and computations. The superior performance with 8K tokens confirms that constrained length can hinder the expression of complete reasoning chains. Therefore, a maximum response length of 8K is identified as the optimal configuration for achieving the best overall performance.

**Analysis of the training entropy.** Fig. 4 compares the evolution of entropy and training rewards throughout the RL training process for both GRPO and our proposed Info-GRPO. Our method demonstrates a faster entropy reduction (in subfigures (a) and (b)), indicating quicker policy convergence. Simultaneously, it achieves a steeper reward increase (subfigures (c) and (d)), signifying more efficient learning. These results confirm that the latent prior in Info-GRPO stabilizes training and accelerates the discovery of high-reward policies, explaining its superior final performance.

**Analysis of the prompt-response correlation.** Based on Eq. (8), a high correlation score is observed only when a trajectory $\tau^i$ is paired with its seed $z^i$, and the score is low for any mismatched pair $(z^j, \tau^i)$ where $j \neq i$. Fig. 1 (c) further visualizes the correlation between different augmented

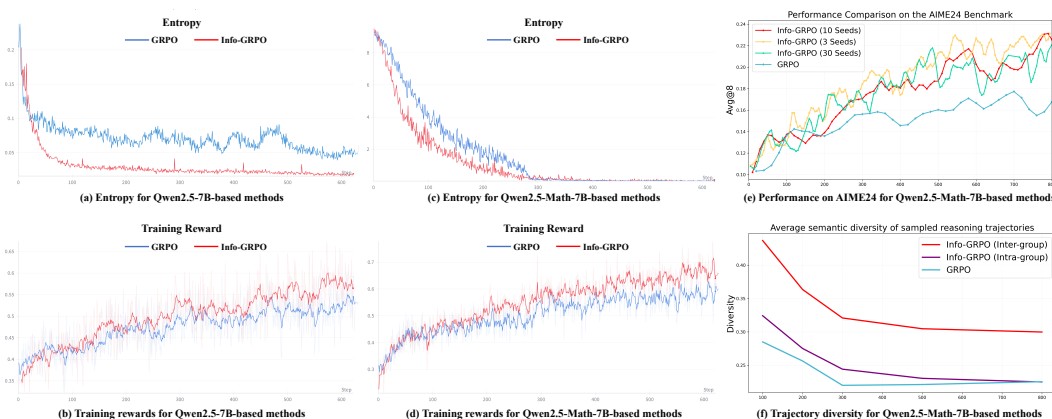

Figure 4: Comprehensive analysis of training dynamics and exploration mechanisms. (a-d) Info-GRPO exhibits entropy reduction while achieving higher training rewards compared to the GRPO baseline. (e) Validation performance on AIME24 confirms that Info-GRPO consistently outperforms GRPO during training, regardless of the seed quantity. (f) Analysis of policy semantic diversity reveals that Info-GRPO (with 10 seeds) maintains high inter-group diversity across conditional seeds and low intra-group diversity within a seed, effectively balancing exploration and exploitation.

prompts (rows) and their corresponding generated responses (columns). The distinct block-diagonal pattern indicates strong intra-group correlation while maintaining clear separation from responses to other prompts, demonstrating a diversity-driven exploration mechanism. This could signify that a trajectory is highly dependent on its seed, enabling a controllable and diverse exploration of the solution space. Consequently, by sampling from the diverse set of seeds, the model can reliably access a wide range of distinct trajectories that it would not have explored otherwise, thus achieving a rich and structured form of exploration. More case analyses are provided in the Appendix A.5.

**Analysis of the number of latent seeds.** Fig. 4(e) further investigates the sensitivity of Info-GRPO to the size of the latent prior space by varying the number of seeds on the AIME 2024 benchmark. The results indicate that Info-GRPO consistently outperforms the GRPO baseline regardless of the seed quantity, demonstrating its robustness. While all settings yield superior performance, the gain appears to stabilize around 10 seeds. This suggests that a moderate number of latent priors is sufficient to capture the necessary diversity of reasoning modes without incurring excessive computational overhead, striking an optimal balance between exploration breadth and training efficiency.

**Analysis of the policy diversity.** To measure diversity between reasoning trajectories, we employ JINA-v2 (Günther et al., 2023) following DRA-GRPO (Chen et al., 2025) to encode reasoning paths and compute the average pairwise Euclidean distance within the sampled groups. As visualized in Fig. 4(f), Info-GRPO maintains significantly higher inter-group diversity across different latent seeds compared to the baseline, while ensuring lower intra-group diversity within the same seed. This confirms that our mutual information objective enforces structured exploration: different latent variables guide the model toward distinct semantic strategies, while the model remains confident within each specific strategy, avoiding the overall policy collapse observed in vanilla GRPO.

## 7 CONCLUSION AND DICUSSION

This paper identifies the fundamental conflict between naive entropy regularization and convergence in language model reasoning, caused by gradient singularities in vast token spaces. We introduce Info-GRPO, an information-theoretic framework that resolves this tension through latent-variable augmentation and mutual information maximization. Extensive experiments demonstrate consistent improvements over GRPO baselines across multiple benchmarks and model architectures. Info-GRPO's superiority stems from its ability to conduct correlation-aware exploration, as evidenced by distinct latent-conditioned reasoning patterns and stable training dynamics. Future work could explore more sophisticated latent variable structures to unlock a richer diversity of reasoning strategies. For instance, employing hierarchically structured latent spaces may allow the model to learn more fine-grained and compositional control over its generative process.

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

# A APPENDIX

This supplementary material details the proposed method and presents additional experimental results. Section A.1 provides additional proof. Section A.2 reviews the related work. Section A.3 shows more comparative results. Section A.4 provides an in-depth analysis of failure scenarios. Section A.5 provides more case analyses. Section A.6 introduces the usage of LLMs.

## A.1 PROOF

**Theorem 1** (Quantitative Bound of the Tail Set). *Let $\pi_\theta(\cdot|s)$ be a policy distribution over a discrete vocabulary $A$ of size $V$. For any probability threshold $\delta \in (0, 1)$, let the tail set $A_\delta$ be defined as the set of tokens whose probability is less than $\delta$:*

$$A_\delta := \{a \in A \mid \pi_\theta(a|s) < \delta\}. \tag{9}$$

*The size of this tail set, $|A_\delta|$, is lower-bounded as follows:*

$$|A_\delta| \geq V - \frac{1}{\delta}. \tag{10}$$

*Proof.* **1. Bounding the size of the head.** First, we partition the vocabulary $A$ into the tail set $A_\delta$ and its complement, the head set $A_\delta^c = A \setminus A_\delta$. By definition, for any token $a \in A_\delta^c$, its probability is bounded below by $\delta$:

$$\forall a \in A_\delta^c, \quad \pi_\theta(a|s) \geq \delta. \tag{11}$$

The total probability mass is constrained by the normalization axiom of probability distributions:

$$\sum_{a \in A} \pi_\theta(a|s) = 1. \tag{12}$$

We can decompose this sum over the head and tail sets:

$$\sum_{a \in A_\delta^c} \pi_\theta(a|s) + \sum_{a \in A_\delta} \pi_\theta(a|s) = 1. \tag{13}$$

Since probabilities are non-negative, $\sum_{a \in A_\delta} \pi_\theta(a|s) \geq 0$. This implies an upper bound on the probability mass contained within the head set:

$$\sum_{a \in A_\delta^c} \pi_\theta(a|s) \leq 1. \tag{14}$$

Combining this with the lower bound on individual token probabilities in the head set, we get:

$$|A_\delta^c| \cdot \delta \leq \sum_{a \in A_\delta^c} \pi_\theta(a|s) \leq 1. \tag{15}$$

From this, we derive a strict upper bound on the size of the head set, $|A_\delta^c|$:

$$|A_\delta^c| \leq \frac{1}{\delta}. \tag{16}$$

This result is critical: it shows that the number of "high-probability" tokens is independent of the vocabulary size $V$ and is solely limited by the chosen threshold $\delta$.

**2. Deriving the lower bound for the tail size.** The size of the tail set is simply the total vocabulary size minus the size of the head set:

$$|A_\delta| = V - |A_\delta^c|. \tag{17}$$

Substituting our upper bound for $|A_\delta^c|$, we obtain the lower bound for the tail size:

$$|A_\delta| \geq V - \frac{1}{\delta}. \tag{18}$$

This proves the first part of the theorem. As $V \to \infty$, the term $1/\delta$ becomes negligible, and thus $|A_\delta| \approx V$. The vast majority of tokens must lie in the tail.

$\square$

**Remark 1.** According to our result, for large $V$, the gradient sum is dominated by at least $V - 1/\delta$ such terms. For instance, in an LLM with $V = 150,000$ and a small probability threshold of $\delta = 10^{-8}$, the head can contain at most $10^8$ tokens (a loose bound), but the tail is guaranteed to contain at least $150,000 - 10^8$, which is nonsensical unless the head is much smaller. A more realistic scenario, if we assume the top 100 tokens hold significant probability mass, we can set $|A_\delta^c| = 100$, implying $\delta \approx 0.01$ at most. Then $|A_\delta| \geq 150,000 - 100 = 149,900$.

**Theorem 2** (Quantitative Bound on Gradient Instability). *Let $\pi_\theta(\cdot|s)$ be a policy over a vocabulary $A$ of size $V$. The entropy gradient can be decomposed into contributions from a head set $A_\delta^c$ and a tail set $A_\delta = \{a \in A \mid \pi_\theta(a|s) < \delta\}$, for any threshold $\delta \in (0,1)$:*

$$\nabla_\theta H(\pi_\theta) = -\underbrace{\sum_{a \in A_\delta^c} \nabla_\theta \pi_\theta(a|s)(1 + \log \pi_\theta(a|s))}_{\mathbf{G}_{head}} - \underbrace{\sum_{a \in A_\delta} \nabla_\theta \pi_\theta(a|s)(1 + \log \pi_\theta(a|s))}_{\mathbf{G}_{tail}}. \quad (19)$$

*The cumulative magnitude of the logarithmic scaling factors from the tail gradient, which we define as the Total Tail Instability (TTI), is lower-bounded and grows linearly with $V$. Specifically:*

$$TTI := \sum_{a \in A_\delta} |1 + \log \pi_\theta(a|s)| \geq \left(V - \frac{1}{\delta}\right)|1 + \log \delta|. \quad (20)$$

*This bound holds provided $V > 1/\delta$, a condition easily met in LLMs.*

*Proof.* **1. Bounding the Magnitude of Each Term.** For any token $a \in A_\delta$, by definition, $0 < \pi_\theta(a|s) < \delta$. Since the logarithm is a monotonically increasing function, this implies $\log \pi_\theta(a|s) < \log \delta$. Therefore, for each term in the TTI sum, we can establish a lower bound on its magnitude:

$$|1 + \log \pi_\theta(a|s)| > |1 + \log \delta|. \quad (21)$$

This holds true because for any small $\delta < 1/e \approx 0.36$, the term $(1 + \log \delta)$ is negative, and its magnitude increases as $\delta$ approaches zero.

**2. Bounding the Number of Terms.** From Theorem 1, we have a tight lower bound on the size of the tail set $|A_\delta|$:

$$|A_\delta| \geq V - \frac{1}{\delta}. \quad (22)$$

By combining these two results, we can lower-bound the Total Tail Instability (TTI):

$$\text{TTI} = \sum_{a \in A_\delta} |1 + \log \pi_\theta(a|s)| > \sum_{a \in A_\delta} |1 + \log \delta| \quad (23)$$

$$= |A_\delta| \cdot |1 + \log \delta| \quad (24)$$

$$\geq \left(V - \frac{1}{\delta}\right)|1 + \log \delta|. \quad (25)$$

This concludes the proof. The TTI, which represents the total amplification of gradient components from the tail, is shown to have a magnitude that scales at least linearly with the vocabulary size $V$. $\qquad \square$

## A.2 RELATED WORK

**Reinforcement Learning for LLM Reasoning.** Reinforcement learning has emerged as a pivotal paradigm for improving the reasoning capabilities of Large Language Models (LLMs) (OpenAI, 2024; 2025; XAI, 2024; Qwen, 2024; Guo et al., 2025). Early work focused on aligning models with human preferences, typically using Reinforcement Learning from Human Feedback (RLHF) Ouyang et al. (2022). This domain could be categorized into online and offline preference optimization. Online methods (Schulman et al., 2017b; Williams, 1992; Shao et al., 2024) generate responses dynamically during training, receiving real-time feedback. In contrast, offline methods (Rafailov et al., 2023; Meng et al., 2024; Ethayarajh et al., 2024) optimize policies using pre-collected preference datasets. Traditional methods like Proximal Policy Optimization (PPO) Schulman et al. (2017b) and REINFORCE Williams (1992) are computationally expensive and suffer

from instability due to the large and discrete action space. A significant advancement is the development of value-model-free methods, such as Group Relative Policy Optimization (GRPO) Shao et al. (2024). It addresses the instability in PPO by using trajectory-level comparisons instead of value networks, thereby reducing computational costs and enhancing the robustness of the training process. Reinforcement Learning with Verifiable Rewards (RLVR) Lambert et al. (2024); Shao et al. (2025) has also emerged as a promising alternative, demonstrating how outcome-based reward signals can enhance reasoning, particularly in domains demanding rigorous logical deduction like mathematics and programming. A growing understanding in the literature Gandhi et al. (2025) indicates that the presence of reasoning behaviors, rather than merely correct answers, is a primary driver of performance gains in RLVR. Recent works such as DRA-GRPO Chen et al. (2025) aim to address this by explicitly incorporating semantic diversity into the reward computation. S-GRPO Dai et al. (2025) improves the performance by encouraging conciseness and incentivizing early thinking termination. Dr. GRPO Liu et al. (2025) removes the length and std normalization terms to avoid the optimization bias in GRPO. DAPO Yu et al. (2025) proposes four effective techniques, such as clip-higher, dynamic sampling, token-level policy gradient loss, and overlong reward shaping. VAPO Yue et al. (2025) further integrates the value model by proposing length-adaptive GAE. ORZ Hu et al. (2025) also utilizes a value model for advantage estimation with the Monte Carlo estimation. Despite the successes of these methods, challenges remain. Notably, GRPO-based approaches struggle with insufficient exploration and the lack of diversity in generated solutions Chen et al. (2025). In this work, we focus on extending GRPO to achieve improvements in exploration and exploitation capabilities.

**Entropy Regularization for Reinforcement Learning.** Entropy regularization has become an essential technique in RL to address issues such as premature convergence and insufficient exploration Hong et al. (2018). Early works focused on entropy as a means to encourage exploration in environments with high uncertainty Mnih et al. (2015; 2016); Haarnoja et al. (2017); Schulman et al. (2017a;b); Haarnoja et al. (2018). In particular, the maximum entropy principle Ziebart et al. (2008); Toussaint (2009) has been used to balance reward maximization with policy stochasticity. It has been extended to language model training, where entropy-based terms are introduced into the reward function to enhance the model's exploratory behaviors during reasoning tasks. Recent works focus on forking tokens, which introduce new reasoning paths to improve the reasoning performance of LLMs when. Wang et al. Wang et al. (2025a) highlight the importance of high-entropy tokens in driving reasoning behavior. FR3E Zheng et al. (2025b) identifies high-uncertainty decision points in reasoning trajectories and builds intermediate feedback by conducting targeted rollouts. Other recent advancements in entropy-based exploration strategies, such as diversity-driven exploration Chen et al. (2025) and the RL with entropy-augmented advantage Cheng et al. (2025), propose solutions by introducing entropy regularization into the advantage function. These methods reinforce exploratory behaviors, allowing LLMs to tackle complex reasoning tasks more effectively. In addition, Skywork-OR1 He et al. (2025) utilizes the appropriate entropy control to mitigate premature convergence and improve test outcomes. 1-shot RLVR Wang et al. (2025b) promotes diverse exploration in outputs by adding an entropy loss with a coefficient to enhance model performance. While effective at both a macroscopic level (preventing overall policy collapse) and a microscopic level (guiding exploration at individual token choices), it is challenging for entropy-based methods to balance exploration and exploitation, since the uncertainty introduced by entropy to promote exploration may weaken the confidence of the model. In this paper, we extend entropy-regularized GRPO with mutual information to address the contradiction between diversity and confidence.

**Mutual Information for Structured Exploration.** In unsupervised learning, Mutual Information (MI) has been widely used to capture dependencies between random variables and improve the diversity of learned representations Hjelm et al. (2018); Poole et al. (2019); Wen et al. (2020); Rakelly et al. (2021); Chen et al. (2024). The power of MI in learning disentangled representations in unsupervised settings is evident in methods like InfoGAN Chen et al. (2016) and InfoVAE Zhao et al. (2017). InfoGAN has disentangled prominent attributes to show its capacity for unsupervised discovery of interpretable concepts. Similarly, InfoVAE addresses limitations of variational autoencoders by incorporating an explicit mutual information constraint between the latent code and the generated data within its loss function. MI also underpins self-supervised contrastive learning, a field that employs the InfoNCE loss Chen et al. (2020); Zhang et al. (2023); Lee et al. (2024) to maximize the similarity between positive sample pairs and minimize the similarity between negative sample pairs by estimating mutual information. In the context of LLMs, mutual information has also been explored for enhancing model reasoning capabilities by ensuring that reasoning steps are not overly deterministic or constrained. For instance, the submodular mutual information used

Table 5: **Comparison of methods on different backbones and benchmarks** ($Avg@8$**, %**)**.**

| Backbone | Method | AIME24 | AIME25 | AMC | MATH500 | Minerva | **Average** |
|---|---|---|---|---|---|---|---|
| *Qwen2.5-7B* | Base Model | 12.08 | 7.50 | 42.77 | 75.50 | 35.06 | 34.58 |
| | GRPO | 20.42 | 10.83 | 56.48 | **80.53** | 37.82 | 41.22 |
| | DAPO | 18.75 | 11.25 | 48.95 | 78.13 | 36.53 | 38.72 |
| | Skywork-OR1 | 20.83 | 7.50 | 57.38 | 78.38 | 38.05 | 40.43 |
| | CLIP-Cov | 15.83 | **12.08** | 51.51 | 79.10 | 38.19 | 39.34 |
| | KL-Cov | 15.40 | 11.25 | **59.94** | 74.05 | 35.94 | 39.32 |
| | Info-GRPO (Ours) | **22.08** | 8.75 | 58.28 | 79.60 | **38.37** | **41.42** |
| *Qwen2.5-Math-7B* | Base Model | 11.67 | 8.33 | 48.64 | 82.38 | 36.40 | 37.48 |
| | GRPO | 21.25 | 11.67 | 66.57 | **86.75** | 38.97 | 45.04 |
| | DAPO | 21.25 | 7.92 | 79.52 | 86.35 | 38.92 | 46.79 |
| | Skywork-OR1 | 22.50 | 13.33 | **81.02** | 84.73 | 38.19 | 47.95 |
| | Dr. GRPO | **29.17** | 9.58 | 58.58 | 79.18 | 34.83 | 42.27 |
| | Info-GRPO (Ours) | 25.00 | **15.83** | 78.46 | 86.63 | **39.34** | **49.05** |
| *DeepSeek-R1-Distill-Qwen-7B* | Base Model | 54.58 | 34.17 | 81.63 | 92.03 | 39.89 | 60.46 |
| | GRPO | 57.92 | 34.17 | 82.38 | 93.63 | 44.16 | 62.45 |
| | DAPO | 58.33 | 37.50 | 82.53 | 93.63 | 44.53 | 63.30 |
| | Skywork-OR1 | 59.58 | 37.92 | 82.08 | 93.73 | 44.12 | 63.48 |
| | Light-R1-7B-DS | 59.10 | 44.30 | 82.83 | 93.55 | 44.35 | 64.83 |
| | Info-GRPO (Ours) | **62.08** | **45.83** | **83.13** | **93.98** | **44.94** | **65.99** |
| *DeepSeek-R1-Distill-Llama-8B* | Base Model | 50.42 | 32.50 | 79.67 | 85.35 | 37.91 | 57.17 |
| | GRPO | 53.75 | 35.42 | 81.02 | 86.23 | 37.22 | 58.73 |
| | DAPO | 51.67 | 36.67 | 80.87 | 91.08 | 37.45 | 59.55 |
| | Info-GRPO (Ours) | **57.92** | **37.50** | **82.23** | **91.45** | **37.96** | **61.41** |

in GRPO-based methods Chen et al. (2025) aims to downweight redundant completions and focus on diverse reasoning outputs. Qian et al. Qian et al. (2025) investigate the reasoning trajectory of large reasoning models from the perspective of information theory. They find that the MI between intermediate representations and the answer arrives at peaks corresponding to tokens that indicate reflection or transition. Moreover, InfoPO Xiao et al. (2025) optimizes the conditional mutual information between responses and preferences given a prompt to avoid the Bradley-Terry assumption. In the context of controllability and randomness in generative models, the mutual information approach is particularly useful for balancing control over the model's output while still allowing for enough randomness to explore diverse reasoning strategies. This motivates the emergence of Info-GRPO in this paper, which explores the combination of unsupervised mutual information maximization and RL techniques. Through maximizing the mutual information conditioned on a new latent variable, our method improves the quality of reasoning and ensures that LLM can handle more complex and abstract reasoning tasks.

## A.3 COMPARATIVE RESULTS

More model series (*e.g.*, DeepSeek-R1-Distill-Llama-8B (Guo et al., 2025)), domains (*e.g.*, Code), and comparative experiments with more recent methods are provided as follows:

**Comparisons with more recent works.** As shown in Table 5, more recent works are compared using the officially released models under the same evaluation configuration as follows:

- *Qwen2.5-7B setting.* For methods such as KL-Cov and Clip-Cov (Cui et al., 2025), which release code but do not provide pretrained models, we train them by using the official codebase and report their best average performance under the standardized evaluation after training convergence. It could be observed that our method performs better on most datasets.

- *Qwen2.5-Math-7B setting.* Dr. GRPO (Liu et al., 2025) is added in Table 5. While Dr. GRPO achieves the highest score on AIME24 (29.17), it performs notably worse on the remaining benchmarks, yielding a much lower overall average (42.27) compared to our method (49.05).

Table 6: **Comparison on Livecodebench based on DeepSeek-R1-Distill-Qwen-7B (Avg@8, %).**

| Metric | $Pass@1$ | $Avg@8$ | $Pass@8$ |
|---|---|---|---|
| Base Model | 36.92 | 36.74 | 48.75 |
| GRPO | 37.63 | 38.13 | 50.18 |
| Info-GRPO (Ours) | **40.86** | **40.14** | **51.97** |

Table 7: **Comparison of method diversity and performance on AIME25 based on Qwen2.5-7B.**

| Method | Diversity (L2 distance) | $Avg@8$ |
|---|---|---|
| GRPO | 0.206 | 10.83 |
| DAPO | 0.190 | 11.25 |
| Skywork-OR1 | 0.239 | 7.50 |
| Info-GRPO (Ours) | 0.287 | 8.75 |

- *DeepSeek-R1-Distill-Qwen-7B setting.* We add comparisons with Light-R1 (Wen et al., 2025). Although Light-R1 (mean 64.83) outperforms other baselines such as GRPO, DAPO, and Skywork-OR1, our method still achieves superior overall results (65.99).

Overall, these additions provide a broader and more up-to-date contextualization of our contributions while ensuring that all quantitative comparisons remain fair and directly comparable.

**Comparisons on more model series.** To further validate the robustness and scalability of our method, we additionally trained GRPO, DAPO, and our Info-GRPO on DeepSeek-R1-Distill-Llama-8B under the same settings. The best-performing results are summarized in Table 5. Info-GRPO consistently achieves the highest accuracy across multiple datasets, demonstrating that our method transfers well beyond Qwen models and can generalize effectively to other model families.

**Comparisons on more domains.** The experiments on the coding domain using Live-CodeBench (Jain et al., 2024) are conducted to demonstrate the versatility of Info-GRPO. Following Skywork-OR1 (He et al., 2025), the models are trained on a 13.7K-sample coding dataset. The DeepSeek-R1-Distill-Qwen-7B is adopted as the base model, and all training hyperparameters are maintained identical to those used in our math experiments. The maximum input token length is $8K$ for consistency with our math benchmarks and to maintain computational efficiency without compromising performance on these tasks. For evaluation, models are evaluated on LiveCodeBench with 279 samples, a challenging, contamination-free benchmark for code generation. Table 6 shows that Info-GRPO consistently outperforms both the base model and GRPO across all metrics, indicating that the improvements brought by our method extend beyond the mathematics domain and apply to broader reasoning tasks.

### A.4 FAILURE ANALYSIS

For the underperformance of our method in Table 1, a deeper investigation is conducted. We have substantially expanded our analysis and added new empirical evidence to clarify the underlying causes as follows:

*Deeper analysis.* AIME25 is a highly demanding mathematical reasoning benchmark, where the headroom for diversity-based improvements is fundamentally constrained by the reasoning capacity of the base model. Although Entropy-regularized methods such as Skywork-OR1 and Info-GRPO generally benefit from increased sample diversity, the effectiveness of diversity depends critically on the quality of the candidate reasoning trajectories that the base model can generate.

When the underlying model's capability is relatively limited, promoting diversity tends to increase the likelihood of generating low-quality or unstable reasoning paths, which can reduce the probability that any of the eight sampled responses is correct.

This hypothesis is further supported by the performance of another recent entropy-regularized method, Skywork-OR1. Although it improves diversity, its performance (7.5) is even lower than Info-GRPO and all other baselines. This serves as an additional indicator that diversity alone does not guarantee improved reasoning accuracy when base-model ability is the limiting factor.

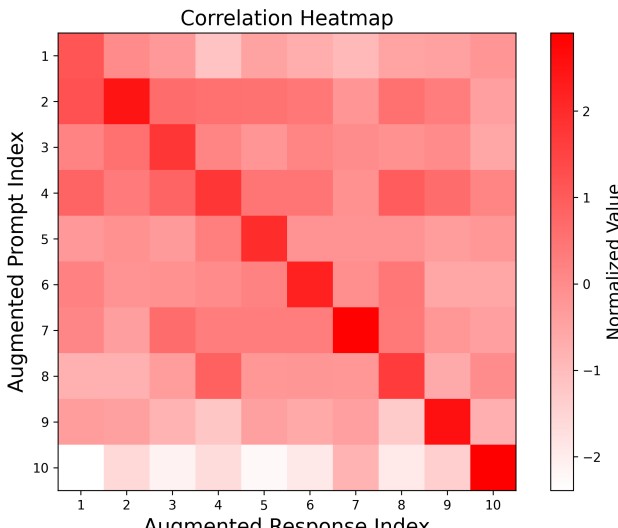

Figure 5: The correlation heatmap for a case. It is generated by our Info-GRPO based on the DeepSeek-R1-Distill-Qwen-7B models.

*Empirical evidence.* Prior works typically measure diversity using token-level entropy, which cannot capture the semantic similarity between whole reasoning trajectories. Following the practice in DRA-GRPO (Chen et al., 2025), we adopt an embedding-based sequence-level diversity. Specifically, the intra-set diversity is computed among the eight sampled solutions per query for all methods. We use JINA-v2 (Günther et al., 2023) (supporting up to 8192 tokens) to obtain embeddings and compute the average pairwise L2 distance within each 8-sample group, and then average this value across the dataset. Table 7 illustrates that Info-GRPO achieves the highest diversity ceiling when averaging over all latent seeds. However, higher diversity does not correlate with higher accuracy on AIME25. Even at its diversity peak, Info-GRPO reaches 8.75, which is higher than Skywork-OR1 but remains below GRPO and DAPO. This pattern is fully consistent with our hypothesis: When the base model struggles to produce stable reasoning trajectories, adding diversity amplifies variance more than it improves correctness.

## A.5 CASE ANALYSIS

It should be noted that in the main text, we only use the augmented prompt for training but not for evaluation. Here, we use different augmented seeds for evaluation just for case analysis. Fig. 5 illustrates a correlation heatmap given a query as follows:

Find the number of ways to place a digit in each cell of a 2x3 grid so that the sum of the two numbers formed by reading left to right is 999, and the sum of the three numbers formed by reading top to bottom is 99. The grid below is an example of such an arrangement because $8 + 991 = 999$ and $9 + 9 + 81 = 99$.

| 0 | 0 | 8 |
|---|---|---|
| 9 | 9 | 1 |

The scores for 8 responses under different prompt seeds are shown in Table 8. It could be observed that responses under different seeds get different scores. Fig. 6 further provides two cases under different prompt seeds. It can be seen that the model adopts completely different approaches to solving the problem, which verifies the effectiveness of our diversity-driven mechanism.

## A.6 USAGE OF LLMS

Large Language Models (LLMs) are employed exclusively for grammar checking in this manuscript. They are not utilized for writing assistance, literature retrieval, or research ideation.

Table 8: The scores for 8 responses under different prompt seeds.

| Seed | No seed | Aug1 | Aug2 | Aug3 | Aug4 | Aug5 | Aug6 | Aug7 | Aug8 | Aug9 | Aug10 |
|------|---------|------|------|------|------|------|------|------|------|------|-------|
| R1 | 1 | 1 | 0 | 1 | 1 | 1 | 0 | 1 | 1 | 1 | 1 |
| R2 | 0 | 1 | 1 | 0 | 0 | 0 | 1 | 1 | 1 | 0 | 1 |
| R3 | 1 | 1 | 1 | 1 | 0 | 0 | 1 | 1 | 1 | 0 | 1 |
| R4 | 0 | 0 | 1 | 0 | 1 | 0 | 0 | 1 | 1 | 1 | 1 |
| R5 | 1 | 1 | 1 | 1 | 1 | 1 | 1 | 1 | 0 | 1 | 1 |
| R6 | 1 | 0 | 1 | 1 | 1 | 0 | 1 | 1 | 0 | 1 | 1 |
| R7 | 1 | 1 | 1 | 1 | 1 | 1 | 1 | 1 | 1 | 1 | 0 |
| R8 | 1 | 0 | 1 | 1 | 1 | 1 | 0 | 1 | 0 | 0 | 0 |

Alright, let me try to figure out how to solve this problem. So, we have a 2x3 grid, right? We need to place digits in each cell such that two conditions are satisfied:
1. The sum of the two numbers formed by reading left to right in each row is 999.
2. The sum of the three numbers formed by reading top to bottom in each column is 99.
......
......
The key steps are:
1. The sum of the top digits $A + B + C = 9$.
2. The sum of the bottom digits $D + E + F = 9$.
3. The sum of the numbers formed by the columns must be 99.
We analyze the columns:
- Column 1: $\text{col1} = D$ if $A = 0$ else $10A + D$
- Column 2: $\text{col2} = E$ if $B = 0$ else $10B + E$
- Column 3: $\text{col3} = F$ if $C = 0$ else $10C + F$
The sum of these columns must be 99:
$$\text{col1} + \text{col2} + \text{col3} = 99$$
After thorough analysis, we find that the only valid grid is when:
- $A = 0$, $B = 9$, $C = 0$
- $D = 9$, $E = 0$, $F = 0$
This results in:
- $\text{col1} = 9$
- $\text{col2} = 90$
- $\text{col3} = 0$
Thus, the sum of the columns is $9 + 90 + 0 = 99$.
Therefore, the number of such grids is:
$$\boxed{1}$$

**(a) The first response under seed aug2 with score 0**

Alright, so I've got this problem where I need to figure out how many ways there are to place a digit in each cell of a 2x3 grid. The conditions are that the sum of the two numbers formed by reading left to right in each row should be 999, and the sum of the three numbers formed by reading top to bottom in each column should be 99.
Let me try to visualize the grid. It's a 2x3 grid, meaning there are 2 rows and 3 columns. Each cell will contain a digit from 0 to 9.
......
......
We find that:
- $c + f = 9$
- $d = b + c + 1$
- $a + d = 9$
- $b + e = 9$
We express the variables in terms of $d$ and $c$:
- $a = 9 - d$
- $b = d - c - 1$
- $e = 10 - d + c$
- $f = 9 - c$
We determine the constraints:
- $d$ ranges from 1 to 9
- For each $d$, $c$ ranges from 0 to $d - 1$
The number of possible pairs $(c, d)$ is the sum from $d = 1$ to $d = 9$ of the number of $c$ values, which is:
$$\sum_{d=1}^{9} d = \frac{9 \times 10}{2} = 45$$
Thus, the number of ways to place the digits in the grid is:
$$\boxed{45}$$

**(b) The first response under seed aug1 with score 1**

Figure 6: Responses under different prompt seeds.

