# OpenReview forum: "Info-GRPO: Training Reasoning Models via Correlation-Aware Exploration"
_ICLR.cc/2026/Conference — Submitted to ICLR 2026_

### Official Review · Reviewer_YdgL · 2025-10-23

**Soundness:** 3
**Presentation:** 3
**Contribution:** 2
**Rating:** 6
**Confidence:** 3

**Summary:**

This paper present Info-GRPO, an information-theoretic framework that reconciles the opposing entropic forces of exploration and convergence by cultivating correlation between the policy and a latent prior. Info-GRPO leverages a contrastive regularization that maximizes the mutual information between latent variables and the policy. Intuitively, by augmenting prompts with latent variables, the model explores a more diverse set of policies that remain correlated with the latent prior, guiding conditional entropy toward convergence. Through this correlation-aware design, Info-GRPO respects the natural entropy reduction during training while enabling more effective exploration.

**Strengths:**

1.  The paper proposes a new perspective by leveraging a contrastive regularization that maximizes the mutual information between latent variables and the policy. By augmenting prompts with latent variables, the model explores a more diverse set of policies that remain correlated with the latent prior, guiding conditional entropy toward convergence. The paper also provides theoretical analysis to support this approach.
2.  The analysis correctly points out that the failure of traditional entropy regularization in LLMs is due to gradient singularities induced by the massive number of tokens in the tail of the distribution.

**Weaknesses:**

1.  Missing baselines: There have been several recent works rethinking entropy regularization, such as KL-Cov and Clip-Cov [1], training on 20% high-entropy tokens [2], and entropy-based advantage shaping [3]. I suggest the authors compare with these baselines for a more up-to-date evaluation.

2.  The evaluation is currently limited to the math domain on Qwen-series models. Extending the experiments to other model series and domains would help assess the robustness of the proposed method.

[1] The entropy mechanism of reinforcement learning for reasoning language models.
[2] Beyond the 80/20 rule: High-entropy minority tokens drive effective reinforcement learning for LLM reasoning.
[3] Reasoning with exploration: An entropy perspective.

**Questions:**

See weakness.

---

> ### Author Response · Authors · 2025-11-20
> **Response to Reviewer YdgL - Part 1**
>
> **Q1.** Comparison with more recent baselines.
>
> **R1.** We sincerely appreciate the reviewer’s insightful suggestion. We have included more comparisons and discussions of the recent entropy-regularization works [1–3]. Additionally, we have compared with other recent baselines such as Dr.GRPO (COLM 2025) [4]  and Light-R1 (ACL 2025) [5], using the officially released models under the same evaluation configuration. Our experiments are conducted primarily on the Qwen2.5 family of models, therefore:
>
> **For methods using the same base models**: More quantitative comparisons with the recent methods providing official code or checkpoints are added:
>
> - Qwen2.5-7B setting: For methods such as KL-Cov and Clip-Cov [1], which release code but do not provide pretrained models, we train them by using the official codebase and report their best average performance under the standardized evaluation after training convergence. It could be observed that our method performs better on most datasets.
>
> - Qwen2.5-Math-7B setting: We compare against Dr. GRPO in Table R5. While Dr. GRPO achieves the highest score on AIME24 (29.17), it performs notably worse on the remaining benchmarks, yielding a much lower overall average (42.27) compared to our method (49.05).
>
> - DeepSeek-R1-Distill-Qwen-7B setting: We add comparisons with Light-R1. Although Light-R1 (mean 64.83) outperforms other baselines such as GRPO, DAPO, and Skywork-OR1, our method still achieves superior overall results (65.99).
>
> **For methods with different architectures or a lack of open-source code**: For methods relying on different model families (e.g., Qwen3-based high-entropy-tokens [2]) or those without publicly available code/models [3], we include explicit citations and discussions without quantitative comparisons.
>
> Overall, these additions provide a broader and more up-to-date contextualization of our contributions while ensuring that all quantitative comparisons remain fair and directly comparable.
>
> **Table R1. Comparison with more noteworthy baselines.**
>
> | **Qwen2.5-7B** | AIME24 | AIME25 | AMC    | MATH500 | Minerva | Average |
> |---------|--------|--------|--------|---------|---------|---------|
> | Base Model| 12.08  | 7.50   | 42.77  | 75.50   | 35.06   | 34.58   |
> | GRPO| 20.42  | 10.83  | 56.48  | **80.53**   | 37.82   | 41.22   |
> | DAPO| 18.75  | 11.25  | 48.95  | 78.13   | 36.53   | 38.72   |
> | Skywork-OR1| 20.83  | 7.50   | 57.38  | 78.38   | 38.05   | 40.43   |
> | **CLIP-Cov [1]**| 15.83  | **12.08**  | 51.51  | 79.10   | 38.19   | 39.34   |
> | **KL-Cov [1]**| 15.40  | 11.25  | **59.94**  | 74.05   | 35.94   | 39.32   |
> | **Info-GRPO (Ours)**|**22.08**|8.75|58.28|79.60|**38.37**|**41.42**|
>
> | **Qwen2.5-Math-7B** | AIME24 | AIME25 | AMC    | MATH500 | Minerva | Average |
> |---------|--------|--------|--------|---------|---------|---------|
> | Base Model| 11.67  | 8.33   | 48.64  | 82.38   | 36.40   | 37.48   |
> | GRPO| 21.25  | 11.67  | 66.57  | **86.75**   | 38.97   | 45.04   |
> | DAPO| 21.25  | 7.92   | 79.52  | 86.35   | 38.92   | 46.79   |
> | Skywork-OR1| 22.50  | 13.33  | **81.02**  | 84.73   | 38.19   | 47.95   |
> | **Dr. GRPO (COLM 2025)**| **29.17**  | 9.58   | 58.58  | 79.18   | 34.83   | 42.27   |
> | **Info-GRPO (Ours)**| 25.00  | **15.83**  | 78.46  | 86.63   | **39.34**   | **49.05**   |
>
> | **DeepSeek-R1-Distill-Qwen-7B** | AIME24 | AIME25 | AMC    | MATH500 | Minerva | Average |
> |---------|--------|--------|--------|---------|---------|---------|
> | Base Model| 54.58  | 34.17  | 81.63  | 92.03   | 39.89   | 60.46   |
> | GRPO| 57.92  | 34.17  | 82.38  | 93.63   | 44.16   | 62.45   |
> | DAPO| 58.33  | 37.50  | 82.53  | 93.63   | 44.53   | 63.30   |
> | Skywork-OR1| 59.58  | 37.92  | 82.08  | 93.73   | 44.12   | 63.48   |
> | **Light-R1-7B-DS (ACL 2025)**   | 59.10  | 44.30  | 82.83  | 93.55   | 44.35   | 64.83   |
> | **Info-GRPO (Ours)**| **62.08**  | **45.83**  | **83.13**  | **93.98**   | **44.94**   | **65.99**   |
>
> [1] The entropy mechanism of reinforcement learning for reasoning language models. arXiv preprint arXiv:2505.22617.
>
> [2] Beyond the 80/20 rule: High-entropy minority tokens drive effective reinforcement learning for LLM reasoning. arXiv preprint arXiv:2506.01939.
>
> [3] Reasoning with exploration: An entropy perspective. arXiv preprint arXiv:2506.14758.
>
> [4] Understanding R1-Zero-Like Training: A Critical Perspective. COLM 2025.
>
> [5] Light-R1: Curriculum SFT, DPO and RL for Long COT from Scratch and Beyond. ACL 2025.
>
> ---

---

> > ### Author Response · Authors · 2025-11-20
> > **Response to Reviewer YdgL - Part 2**
> >
> > **Q2.** Extending the experiments to other model series.
> >
> > **R2.** Thank you for the insightful comment. To further validate the robustness and scalability of our method, we additionally trained GRPO, DAPO, and our Info-GRPO on DeepSeek-R1-Distill-Llama-8B under the same settings. The best-performing results are summarized in the table below. Info-GRPO consistently achieves the highest accuracy across multiple datasets, demonstrating that our method transfers well beyond Qwen models and can generalize effectively to other model families.
> >
> > **Table R2. Comparison of methods on different benchmarks based on DeepSeek-R1-Distill-Llama-8B (Avg@8, %).**
> >
> > | DeepSeek-R1-Distill-Llama-8B | AIME24 | AIME25 | AMC    | MATH500 | Minerva | Average |
> > |---------|--------|--------|--------|---------|---------|---------|
> > | Base Model| 50.42  | 32.50  | 79.67  | 85.35   | 37.91   | 57.17   |
> > | GRPO| 53.75  | 35.42  | 81.02  | 86.23   | 37.22   | 58.73   |
> > | DAPO  | 51.67  | 36.67  | 80.87  | 91.08   | 37.45   | 59.55   |
> > | **Info-GRPO (Ours)**| **57.92**  | **37.50**  | **82.23**  | **91.45**   | **37.96**   | **61.41** |
> >
> >
> > ---
> >
> > **Q3.** Extending the experiments to other domains.
> >
> > **R3.** We agree with the reviewer that verifying the generalization capability of our method across different domains is essential. We have conducted additional experiments on the coding domain using LiveCodeBench [1] to demonstrate the versatility of Info-GRPO.
> >
> > **Training details**: Following Skywork-OR1 (He et al., 2025), we train the models on a 13.7K-sample coding dataset (2.7K from LeetCode and 11K from TACO). The DeepSeek-R1-Distill-Qwen-7B is adopted as the base model, and all training hyperparameters are maintained identical to those used in our math experiments. The maximum input token length is 8k for consistency with our math benchmarks and to maintain computational efficiency without compromising performance on these tasks.
> >
> > **Evaluation**: Models were evaluated on LiveCodeBench (279 samples), a challenging, contamination-free benchmark for code generation.
> >
> > **Results**: The results are summarized in Table R1. Info-GRPO consistently outperforms both the base model and GRPO across all metrics, indicating that the improvements brought by our method extend beyond the mathematics domain and apply to broader reasoning tasks.
> >
> > **Table R3. Comparison of different methods on Livecodebench based on DeepSeek-R1-Distill-Qwen-7B.**
> >
> > | DeepSeek-R1-Distill-Qwen-7B | Pass@1 | Avg@8 | Pass@8 |
> > |---|---|---|---|
> > | Base Model | 36.92  | 36.74  | 48.75  |
> > | GRPO | 37.63  | 38.13  | 50.18  |
> > | **Info-GRPO** | **40.86**  | **40.14**  | **51.97** |
> >
> > [1] Livecodebench: Holistic and contamination free evaluation of large language models for code. arXiv preprint arXiv:2403.07974, 2024.
> >
> > ---

---

> > > ### Comment · Reviewer_YdgL · 2025-11-26
> > >
> > > Thanks for the authors' response; my review remains the same (6: marginally above the acceptance threshold).

---

> > > > ### Author Response · Authors · 2025-11-26
> > > > **Appreciation for the positive assessment**
> > > >
> > > > Dear Reviewer,
> > > >
> > > > We sincerely thank you for your thoughtful feedback and for maintaining a positive overall assessment of our submission. Your comments prompted us to further validate the effectiveness and robustness of the proposed method through additional experiments, and we truly appreciate your acknowledgment of these efforts. Thank you again for your valuable time and kind help.
> > > >
> > > > Sincerely,
> > > >
> > > > The Authors

---

### Official Review · Reviewer_vQ2H · 2025-11-03

**Soundness:** 3
**Presentation:** 3
**Contribution:** 2
**Rating:** 4
**Confidence:** 4

**Summary:**

Reinforcement Learning with Verifiable Rewards (RLVR) is a primary training paradigm used to post-train modern-day Large Language Models (LLMs). Despite its popularity, RLVR is still plagued with several limitations, one of them being its limited ability to encourage exploration. While entropy regularization is commonly used in RL to offset the policy collapse caused by the lack of policy diversity, its optimization objective, which drives the policy towards a higher entropy region, inherently goes against the "natural optimization trajectory." Therefore, this work proposes Info-GRPO, a novel training framework that attains a balance between exploration and convergence without utilizing opposing objectives. Info-GRPO augments prompts with latent variables and maximizes the mutual information between them, such that a diverse set of reasoning strategies can be sought throughout the training process. The experimental results highlight the limitations of entropy regularization-based exploration in RLVR for LLMs and showcase the effectiveness of Info-GRPO across diverse benchmarks and models.

**Strengths:**

- The proposed method is well-motivated and technically sound. The authors show that entropy regularization is ineffective in RL-based LLM post-training due to the sheer vocabulary size, and then propose a novel framework that avoids policy collapse without relying on entropy regularization.

- The proposed method is simple and intuitive (I meant this as a compliment). Its implementation is easy to follow and can be reproduced at ease.

- Although the performance improvement is marginal in some cases, it is consistently observed across diverse experimental scenarios (models, datasets).

- The authors present thorough ablation studies that analyze the effectiveness of each technical component.

**Weaknesses:**

- I believe it is standard practice nowadays to verify the results of LLM RL on two different domains - coding and mathematics. I kindly ask the authors to additionally share the results in the coding domain to show that Info-GRPO can be used for various problem-solving tasks.

- Although the authors present some case analysis in section A.3, I would appreciate a more structured (or quantitative, if possible) analysis, which clearly evidences that Info-GRPO obtains improved performance by seeking more diverse reasoning trajectories. I believe such an analysis would further set this work apart from other countless GRPO variants. The authors' explanation behind why Info-GRPO does not obtain a high Pass@1 score touches on this (lines 400-403), but this is also a mere guess based on a piece of experiment.

- Also, I found it quite intriguing that randomizing the initial seeds alone could induce sufficient diversity in reasoning trajectories (especially as training progresses). Thus, explicitly analyzing how reasoning trajectories across different seeds evolve over time would aid my understanding (similar to the above point). Also, what happens if a greater/smaller number of latent priors are used to condition the policy?

- I acknowledge that this work does not have to be compared against all variants of GRPO because not all of them share the same goal. Nonetheless, it would be meaningful to add a few more noteworthy baselines to the results (e.g., Dr. GRPO, GSPO).

**Questions:**

Please refer to the Weaknesses section.

---

> ### Author Response · Authors · 2025-11-20
> **Response to Reviewer vQ2H - Part 1**
>
> **Q1.** The results in the coding domain.
>
> **R1.** We appreciate the reviewer’s insightful suggestion that verifying the generalization capability of our method across different domains is essential. We have conducted additional experiments on the coding domain using LiveCodeBench [1] to demonstrate the versatility of Info-GRPO.
>
> **Training details**: Following Skywork-OR1 (He et al., 2025), we train the models on a 13.7K-sample coding dataset (2.7K from LeetCode and 11K from TACO). The DeepSeek-R1-Distill-Qwen-7B is adopted as the base model, and all training hyperparameters are maintained identical to those used in our math experiments. The maximum input token length is 8k for consistency with our math benchmarks and to maintain computational efficiency without compromising performance on these tasks.
>
> **Evaluation**: Models were evaluated on LiveCodeBench (279 samples), a challenging, contamination-free benchmark for code generation.
>
> **Results**: The results are summarized in Table R1. Info-GRPO consistently outperforms both the base model and GRPO across all metrics, indicating that the improvements brought by our method extend beyond the mathematics domain and apply to broader reasoning tasks.
>
> **Table R1. Comparison of different methods on Livecodebench based on DeepSeek-R1-Distill-Qwen-7B.**
>
> | DeepSeek-R1-Distill-Qwen-7B | Pass@1 | Avg@8 | Pass@8 |
> |---|---|---|---|
> | Base Model | 36.92  | 36.74  | 48.75  |
> | GRPO | 37.63  | 38.13  | 50.18  |
> | **Info-GRPO** | **40.86**  | **40.14**  | **51.97** |
>
> [1] Livecodebench: Holistic and contamination free evaluation of large language models for code. arXiv preprint arXiv:2403.07974, 2024.
>
> ---
>
> **Q2.**: Provide a structured/quantitative analysis evidencing that Info-GRPO obtains improved performance by seeking diverse reasoning trajectories.
>
>
> **R2.** We appreciate the suggestion. We provide two quantitative analyses to evidence the role of diversity.
>
> **1. Performance Consistency (Avg@8) as a Diversity Proxy.**
> Info-GRPO is evaluated using standard sampling (without latent seeds). While Pass@1 is noisy on small test sets (30 samples), Avg@8 is a robust indicator of the solution distribution.
>
> Evidence: Info-GRPO consistently dominates Avg@8 (e.g., +4.16% over GRPO). A high Avg@8 under standard sampling proves the policy has internalized a broader distribution of valid solutions, rather than collapsing to a single path.
>
> **2. Semantic Diversity Analysis.**
> To quantify "reasoning trajectory diversity," we calculated the pairwise cosine distance of generated responses using JINA Embeddings v2 during training, as shown in Fig. 4(f) in the revised paper. We observed a clear hierarchy in diversity scores:
>
>
> $$\text{Diversity(Info-GRPO Seeded)} \gg \text{Diversity(Info-GRPO Standard)} \geq \text{Diversity(Baseline GRPO)}$$
>
> Conclusion:
>
>  - Training: Seed-augmented sampling (used only during training) forces high-diversity exploration ($\gg$).
>
> - Inference: Standard sampling (used during evaluation) retains diversity levels comparable to the baseline, yet achieves significantly higher accuracy.
> This confirms that seeds act as "diversity scaffolding" during training. The model successfully internalizes this experience, resulting in a superior policy that performs well without needing external forcing at inference time.
>
> **Table R2. Diversity comparison of different methods on AIME 2024 across training steps based on Qwen2.5-Math-7B.**
>
> | Method           | 100   | 200   | 300   | 500   | 800   | 1600  |
> |------------------|-------|-------|-------|-------|-------|-------|
> | GRPO             | 0.28  | 0.26  | 0.22  | 0.22  | 0.23  | 0.21  |
> | Info-GRPO (Ours) | **0.44**  | **0.36**  | **0.32**  | **0.30**  | **0.30**  | **0.29**  |
>
>
>
> ---

---

> > ### Author Response · Authors · 2025-11-20
> > **Response to Reviewer vQ2H - Part 2**
> >
> > **Q3.** Analyze how reasoning trajectories across different seeds evolve over time to explain why random seeds induce diversity.
> >
> > **R3.** We appreciate the reviewer's curiosity regarding this mechanism. To visualize this evolution, we tracked the semantic diversity (measured by JINA embeddings cosine distance) of the Qwen2.5-Math-7B model on AIME 2024 throughout the training process. We distinguish between:
> >
> > - Intra-group Diversity: Diversity among responses generated from the same latent seed.
> >
> > - Inter-group Diversity: Diversity among responses generated from different latent seed.
> >
> > **Analysis of Evolution**:
> >
> > Emergence of Structured Exploration: As shown in Table R3, Inter-group diversity is consistently higher than Intra-group diversity throughout the entire training process. This confirms that the polices distinguish between seeds.
> >
> > Resistance to Mode Collapse: As training progresses (from Step 100 to 1600), general diversity naturally decays as the model converges on high-reward solutions (Intra-group drops from 0.32 to 0.22). Crucially, however, the Inter-group diversity stabilizes at a significantly higher level (0.29).
> >
> > This gap indicates that Info-GRPO successfully prevents the policy from collapsing into a single deterministic mode. Even at convergence, the latent seeds continue to serve as anchors for distinct, high-quality reasoning strategies, validating that the random seeds function not merely as noise but as structural scaffolding for diversity.
> >
> > **Table R3. Comparison of intra- and inter-group diversity on AIME 2024 across training steps based on Qwen2.5-Math-7B.**
> >
> > | Diversity   | 100   | 200   | 300   | 500   | 800   | 1600  |
> > |-------------|-------|-------|-------|-------|-------|-------|
> > | Intra-group | 0.32  | 0.28  | 0.24  | 0.23  | 0.22  | 0.22  |
> > | Inter-group | **0.44**  | **0.36**  | **0.32**  | **0.30**  | **0.30**  | **0.29**  |
> >
> > ---
> >
> > **Q4.** Analysis for the number of latent priors.
> >
> > **R4.** We conducted an ablation study on the AIME 2024 benchmark using Qwen2.5-Math-7B to investigate the impact of the number of latent seeds ($|Z|$).
> >
> > Detriment of Excessive Seeds ($|Z|=30$): Increasing the number of seeds to 30 leads to a notable performance drop (Pass@1 drops to 26.67%). This suggests that an excessive number of latent priors might be difficult for the model to establish strong, distinct correlations for so many seeds simultaneously.
> >
> > Robustness of Small/Medium Seeds ($|Z|=3, 10$): The model performs robustly with a smaller number of seeds. $|Z|=3$ achieves identical Pass@1 and Avg@8 to $|Z|=10$, indicating that even a few distinct reasoning modes are sufficient to capture the benefits of diversity for this task.
> >
> > Selection Rationale: We selected $|Z|=10$ as a balanced choice to provide sufficient capacity for diverse strategies (potentially capturing more reasoning patterns than 3 seeds) while maintaining training stability.
> >
> > **Table R4. Comparison of methods with different numbers of seeds on AIME24 based on Qwen2.5-Math-7B.**
> >
> > | Number of Seeds | Pass@1 | Avg@8  | Pass@8 | Diversity |
> > |-------------------------|--------|--------|--------|-----------|
> > | 3                       | 30.00  | 25.00  | 40.00  | 0.28 |
> > | 30                      | 26.67  | 23.33  | 30.00  | 0.31 |
> > | 10 (Used in the paper)  | 30.00  | 25.00  | 36.67  | 0.30 |
> >
> > ---

---

> > > ### Author Response · Authors · 2025-11-20
> > > **Response to Reviewer vQ2H - Part 3**
> > >
> > > **Q5.** Comparison with more noteworthy baselines.
> > >
> > > **R5.** We sincerely thank the reviewer for this constructive suggestion. We have included more discussions and comparisons to the noteworthy baselines, including Dr. GRPO [1] and GSPO [2]. Additionally, we have compared with other recent baselines such as Light-R1 (ACL 2025) [3] and Covariance-based methods (Clip-Cov and KL-Cov) [4], using the officially released models or code under the same evaluation configuration. Our experiments are conducted primarily on the Qwen2.5 family of models, therefore:
> > >
> > > **For methods using the same base models**: More quantitative comparisons with the recent methods providing official code or checkpoints are added:
> > >
> > > - Qwen2.5-Math-7B setting: We compare against Dr. GRPO (COLM 2025) in Table R5. While Dr. GRPO achieves the highest score on AIME24 (29.17), it performs notably worse on the remaining benchmarks, yielding a much lower overall average (42.27) compared to our method (49.05).
> > >
> > > - DeepSeek-R1-Distill-Qwen-7B setting: We add comparisons with Light-R1 (ACL 2025) [3]. Although Light-R1 (mean 64.83) outperforms other baselines such as GRPO, DAPO, and Skywork-OR1, our method still achieves superior overall results (65.99).
> > >
> > > - Qwen2.5-7B setting: For methods such as KL-Cov and Clip-Cov [1], which release code but do not provide pretrained models, we train them by using the official codebase and report their best average performance under the standardized evaluation after training convergence. It could be observed that our method performs better on most datasets.
> > >
> > > **For methods with different experimental goals or architectures**, such as GSPO, which is specifically designed for Mixture-of-Experts (MoE) RL training and evaluated on a much larger Qwen3-30B-A3B-Base, we include explicit citations and discussion but do not report direct comparisons due to incompatibility of settings.
> > >
> > > Overall, these additions provide a broader and more up-to-date contextualization of our contributions while ensuring that all quantitative comparisons remain fair and directly comparable.
> > >
> > > **Table R5. Comparison with more noteworthy baselines.**
> > >
> > > | **Qwen2.5-Math-7B** | AIME24 | AIME25 | AMC    | MATH500 | Minerva | Average |
> > > |---------|--------|--------|--------|---------|---------|---------|
> > > | Base Model| 11.67  | 8.33   | 48.64  | 82.38   | 36.40   | 37.48   |
> > > | GRPO| 21.25  | 11.67  | 66.57  | **86.75**   | 38.97   | 45.04   |
> > > | DAPO| 21.25  | 7.92   | 79.52  | 86.35   | 38.92   | 46.79   |
> > > | Skywork-OR1| 22.50  | 13.33  | **81.02**  | 84.73   | 38.19   | 47.95   |
> > > | **Dr. GRPO (COLM 2025)**| **29.17**  | 9.58   | 58.58  | 79.18   | 34.83   | 42.27   |
> > > | **Info-GRPO (Ours)**| 25.00  | **15.83**  | 78.46  | 86.63   | **39.34**   | **49.05**   |
> > >
> > > | **DeepSeek-R1-Distill-Qwen-7B** | AIME24 | AIME25 | AMC    | MATH500 | Minerva | Average |
> > > |---------|--------|--------|--------|---------|---------|---------|
> > > | Base Model| 54.58  | 34.17  | 81.63  | 92.03   | 39.89   | 60.46   |
> > > | GRPO| 57.92  | 34.17  | 82.38  | 93.63   | 44.16   | 62.45   |
> > > | DAPO| 58.33  | 37.50  | 82.53  | 93.63   | 44.53   | 63.30   |
> > > | Skywork-OR1| 59.58  | 37.92  | 82.08  | 93.73   | 44.12   | 63.48   |
> > > | **Light-R1-7B-DS (ACL 2025)**   | 59.10  | 44.30  | 82.83  | 93.55   | 44.35   | 64.83   |
> > > | **Info-GRPO (Ours)**| **62.08**  | **45.83**  | **83.13**  | **93.98**   | **44.94**   | **65.99**   |
> > >
> > > | **Qwen2.5-7B** | AIME24 | AIME25 | AMC    | MATH500 | Minerva | Average |
> > > |---------|--------|--------|--------|---------|---------|---------|
> > > | Base Model| 12.08  | 7.50   | 42.77  | 75.50   | 35.06   | 34.58   |
> > > | GRPO| 20.42  | 10.83  | 56.48  | **80.53**   | 37.82   | 41.22   |
> > > | DAPO| 18.75  | 11.25  | 48.95  | 78.13   | 36.53   | 38.72   |
> > > | Skywork-OR1| 20.83  | 7.50   | 57.38  | 78.38   | 38.05   | 40.43   |
> > > | **CLIP-Cov [4]**| 15.83  | **12.08**  | 51.51  | 79.10   | 38.19   | 39.34   |
> > > | **KL-Cov [4]**| 15.40  | 11.25  | **59.94**  | 74.05   | 35.94   | 39.32   |
> > > | **Info-GRPO (Ours)**|**22.08**|8.75|58.28|79.60|**38.37**|**41.42**|
> > >
> > > [1] Understanding R1-Zero-Like Training: A Critical Perspective. COLM 2025.
> > >
> > > [2] GSPO：Group Sequence Policy Optimization. arXiv preprint arXiv:2507.18071.
> > >
> > > [3] Light-R1: Curriculum SFT, DPO and RL for Long COT from Scratch and Beyond. ACL 2025.
> > >
> > > [4] The entropy mechanism of reinforcement learning for reasoning language models. arXiv preprint arXiv:2505.22617.
> > >
> > > ---

---

### Official Review · Reviewer_HA2y · 2025-11-03

**Soundness:** 3
**Presentation:** 2
**Contribution:** 3
**Rating:** 4
**Confidence:** 4

**Summary:**

This paper examines the tradeoff in RL (GRPO) conflicting objectives with respect to diversity / entropy (that favour exploration) versus the collapse in singular high-entropy configurations.  The paper's analysis conclude that this an inherent modelling challenge due to the vast action space of almost uniform low probabilities, due to the huge vocabulary (token) space.

The authors then construct a regularisation framework to apply to GRPO that conditions on a set of latent variables to add additional constraints to the cost function

**Strengths:**

* I really enjoyed the cause analysis of the failure of high entropy states as attributed to the long vocabulary $|V|$ tail.
* Plots in Fig 4 are very nice to motivate the output of Info-GRPO (but it takes a lot of room, there should be a better way to illustrate the same but with less space).

**Weaknesses:**

* Using latent codes to condition on as a regularisation technique is not necessarily well-motivated by your analysis; there are other forms of embodiment that I can think of.  Help your audience understand **why** latent codes directly follow from your causal analysis.  That is, I found the storytelling of the solutioning not compelling and disconnected from the motivating analysis.
* I find some of the claims of underperformance of Info-GPRO too shallow.  Some of the analysis contributes too little to explanation, without any further empirical investigation to back up its hypotheses. unlike the motivation for the method (which I felt was strong).  More substantial analysis would help to strengthen their and make the paper more uniformly justified.
* I found the choice of evaluation metrics Why choose K as 8 for Pass@K metric?  It's not clear whether this is adopting prior evaluation standards set by prior work.  Help defend that these values aren't cherry picked (Tables 2-4 round percentage values are particularly suspect, given the supposed large sample sizes).  I might just have favoured the reporting of Pass@1 and Avg@8 (or Majority@{K=8}) as others in this line report.
  * Related, why isn't some form of a MRR metric chosen?  Having some favouring of an earlier rank carrying better value makes sense in my opinion.
* It seems you are using $\LaTeX$ for typesetting.  Please fix your use of quotation marks accordingly: ``x'' versus ''x''
*While I agree that the related work is comprehensive (but a bit unfocused), factoring in some parts of this to the main paper is essential: your use of the Appendix A.2 I feel is cheating space requirements (I have to mark you down for this; I feel this is an unfair usage of space)

**Questions:**

* Why / how did you choose a value of 16 for the number of samples per prompt?
* Are the $\epsilon$ values for clipping in DAPO the standard ones?

---

> ### Author Response · Authors · 2025-11-20
> **Response to Reviewer HA2y - Part 1**
>
> **Q1. Fig. 4 should be illustrated with less space.**
>
> **R1.** We sincerely thank the reviewer for the positive feedback regarding the plots. Following this insightful suggestion, we have revised the layout of Fig. 4 to present the information more compactly in the revised manuscript.
>
> ---
>
> **Q2.** Why latent codes directly follow from your causal analysis?
>
> **R2.** Thank you for this constructive suggestion. We have revised Section 5.1 to clarify that Info-GRPO is not just an alternative regularizer, but the direct mathematical solution to the stability problem identified in our analysis. The logic proceeds as follows:
>
> **1. Asymmetric Stability**: As detailed in Section 4, maximizing entropy ($\max H(\pi)$) is numerically unstable due to the singularity trap in the long tail. However, as noted in Remark 3, minimizing entropy is numerically stable and robust.
>
> **2. MI as a Robust Extension**: Mutual Information (MI) is the most direct information-theoretic extension of entropy, defined as $I(\tau, z) = H(\pi) - H(\pi|z)$. Our objective function (Eq. 5) is essentially a generalized implementation of this MI principle. It leverages the conditional entropy minimization term ($-H(\pi|z)$) for stable convergence, while using the coefficient $\alpha$ to dampen the unstable marginal entropy term.
>
> **3. The Role of Latent Codes**: By conditioning on a latent variable $z$, we effectively shift the exploration mechanism from the unstable maximization of global entropy to the stable minimization of conditional entropy. The latent variable $z$ is therefore not an arbitrary addition, but the necessary structural component that allows the model to maintain high confidence (low entropy) locally for each $z$, while achieving diversity globally.
>
> ---
>
> **Q3.** More substantial analysis for underperformance.
>
> **R3.** We have substantially expanded our analysis and added new empirical evidence to clarify the underlying causes.
>
> **1. Deeper analysis**:
> AIME25 is a highly demanding mathematical reasoning benchmark, where the headroom for diversity-based improvements is fundamentally constrained by the reasoning capacity of the base model. Although Entropy-regularized methods such as Skywork-OR1 and Info-GRPO generally benefit from increased sample diversity, the effectiveness of diversity depends critically on the quality of the candidate reasoning trajectories that the base model can generate.
>
> When the underlying model’s capability is relatively limited, promoting diversity tends to increase the likelihood of generating low-quality or unstable reasoning paths, which can reduce the probability that any of the eight sampled responses is correct.
>
> This hypothesis is further supported by the performance of another recent entropy-regularized method, Skywork-OR1. Although it improves diversity, its performance (7.5) is even lower than Info-GRPO and all other baselines. This serves as an additional indicator that diversity alone does not guarantee improved reasoning accuracy when base-model ability is the limiting factor.
>
> **2. Empirical evidence**:
> Prior works typically measure diversity using token-level entropy, which cannot capture the semantic similarity between whole reasoning trajectories. Following the practice in DRA-GRPO [1], we adopt an embedding-based sequence-level diversity. Specifically, we compute the intra-set diversity among the eight sampled solutions per query for all methods. We use jina-embeddings-v2-base-en (supporting up to 8192 tokens) to obtain embeddings and compute the average pairwise L2 distance within each 8-sample group, and then average this value across the dataset.
>
>
> **Table R1. Comparison of method diversity and performance on AIME25 based on Qwen2.5-7B.**
>
> |Method|Diversity (L2 distance)|Avg@8|
> | - | - | - |
> |GRPO|0.206|10.83|
> |DAPO|0.190|11.25|
> |Skywork-OR1|0.239|7.50|
> |Info-GRPO (ours)|0.287|8.75|
>
> Info-GRPO achieves the highest diversity ceiling when averaging over all latent seeds. However, higher diversity does not correlate with higher accuracy on AIME25. Even at its diversity peak, Info-GRPO reaches 8.75, which is higher than Skywork-OR1 but remains below GRPO and DAPO. This pattern is fully consistent with our hypothesis: **When the base model struggles to produce stable reasoning trajectories, adding diversity amplifies variance more than it improves correctness.**
>
> We have incorporated this analysis into the revised manuscript and believe it provides a more complete and empirically grounded justification of the observed results.
>
> [1] DRA-GRPO: Exploring Diversity-Aware Reward Adjustment for R1-Zero-Like Training of Large Language Models arXiv preprint arXiv:2505.09655, 2025.
>
> ---

---

> > ### Author Response · Authors · 2025-11-20
> > **Response to Reviewer HA2y - Part 2**
> >
> > **Q4.** Why choose $K$as 8 for Pass@$K$ metric?
> >
> > **R4.** We appreciate the reviewer's scrutiny regarding our evaluation setup.
> >
> > **1. Rationale for $K$=8 and Metric Selection.**
> > There is currently no single unified standard for $K$ in the reasoning literature, with different works adopting different values, often based on computational constraints:
> >
> > **Varied Standards**: Recent works employ a wide range of settings. For instance, Beyond the 80/20 Rule [1] reports Avg@16. DAPO [2] reports Avg@32, while Reasoning with Exploration [3] uses different $K$ values across benchmarks. Pass@K Policy Optimization [4] also provides a detailed analysis of how metrics vary with $K$.
> >
> > **Our Choice (Balance & Efficiency)**: Since multiple studies have shown that Pass@K typically increases monotonically with $K$ when $K$ is not too large, we fixed $K=8$ to align with the ablation settings commonly used in works like Skywork-OR1 [5]. This value serves as a reasonable balance between computational efficiency and estimation stability.
> >
> > **Robustness**: We prioritize Avg@8 (expected Pass@1) in our reporting, as it is widely recognized as a more robust estimator of policy quality than a single-sample Pass@1.
> >
> > **2. Defense Against Cherry-Picking.**
> > To ensure transparency and facilitate direct comparison, we do not rely on a single metric.
> >
> > **Multiple Metrics**: As shown in the paper, we report Pass@1, Pass@8, and Avg@8 to ensure that our method's performance is verifiable under both single-attempt (Pass@1) and multi-attempt (Pass@8) regimes.
> >
> > **Revision Update**: Following your suggestion to favor Pass@1 and Avg@8, we have expanded Table 1, which presents the results for avg@8, to include all Pass@1 results in Table R2, thus ensuring a more comprehensive comparison with other methods. It could be observed that the Pass@1 results exhibit an overall trend similar to Avg@8, with expected minor variations due to sampling randomness.
> >
> > **Table R2. Comparison of methods on different backbones and benchmarks (Pass@1, %).**
> >
> > | Qwen2.5-7B  | AIME24 | AIME25 | AMC | MATH500 | Minerva | Average  |
> > |---|---|---|---|---|---|---|
> > | Base Model | 13.33  | 6.67  | 40.96  | 75.40  | 35.66  | 34.40   |
> > | GRPO | 16.67  | 13.33  | 57.83  | 77.60  | 37.50  | 40.59   |
> > | DAPO | 13.33  | **16.67**  | 51.81  | 78.60  | 35.66  | 39.21   |
> > | Skywork-OR1 | 16.67  | 10.00  | 57.83  | 77.80  | 38.24  | 40.11   |
> > | Info-GRPO |**16.67**| 13.33 |**59.04**|**79.40**|**40.07**|**41.70**|
> >
> > | Qwen2.5-Math-7B | AIME24 | AIME25 | AMC | MATH500 | Minerva | Average  |
> > |---|---|---|---|---|---|---|
> > | Base Model | 13.33  | 6.67  | 49.40  | 82.20  | 35.29  | 37.38   |
> > | GRPO | 23.33  | 10.00  | 69.88  | 86.80  | 37.13  | 45.43   |
> > | DAPO | 26.67  | 6.67  | 77.11  | 86.60  | 36.03  | 46.62   |
> > | Skywork-OR1 | 26.67  | 16.67  | **80.72**  | 84.20  | 37.13  | 49.08   |
> > | Info-GRPO | **30.00**|**16.67**| 78.31| **86.80**| **38.97**| **50.15**|
> >
> > | DeepSeek-R1-Distill-Qwen-7B | AIME24 | AIME25 | AMC | MATH500 | Minerva | Average  |
> > |---|---|---|---|---|---|---|
> > | Base Model | 53.33  | 33.33  | 79.52  | 92.22  | 39.34  | 59.55   |
> > | GRPO | **66.67**  | 40.00  | 79.52  | 95.00  | 42.65  | 64.77   |
> > | DAPO | 63.33  | 36.67  | 80.72  | 93.00  | 44.12  | 63.57   |
> > | Skywork-OR1 | 63.33  | 40.00  | 81.93  | **95.20**  | 42.28  | 64.55   |
> > | Info-GRPO | 56.67| **50.00** | **84.34** | 94.80 | **46.32**|**66.43**|
> >
> > [1] Beyond the 80/20 rule: High-entropy minority tokens drive effective reinforcement learning for LLM reasoning. arXiv preprint arXiv:2506.01939, 2025.
> >
> > [2] DAPO: An open-source LLM reinforcement learning system at scale. arXiv preprint arXiv:2503.14476, 2025.
> >
> > [3] Reasoning with exploration: An entropy perspective. arXiv preprint arXiv:2506.14758, 2025.
> >
> > [4] Pass@ K Policy Optimization: Solving Harder Reinforcement Learning Problems. arXiv preprint arXiv:2505.15201, 2025.
> >
> > [5] Skywork open reasoner 1 technical report. arXiv preprint arXiv:2505.22312, 2025.
> >
> > ---
> >
> >
> > **Q5.** Are the round percentage values (e.g., 30.00%, 80.00%) suspect?
> >
> > **R5.** The reviewer correctly noted that some values in Tables 2-4 (e.g., 30.00%, 53.33%, 80.00%) appear surprisingly round. This is not due to rounding or cherry-picking, but is a mathematical necessity given the small size of the AIME test sets. As detailed in Section 6.2 BENCHMARKS AND METRICS, the AIME 2024 and AIME 2025 test sets contain exactly 30 samples each. With a denominator of 30, every single problem solved contributes exactly $1/30 \approx 3.333\%$ to the score. Consequently, most accuracy scores are multiples of $3.33\%$ as follows:
> >
> > $9/30 = 30.00\%$
> >
> > $16/30 = 53.333...\% \rightarrow 53.33\%$
> >
> > $24/30 = 80.00\%$
> >
> > Therefore, these values faithfully reflect the discrete nature of the AIME benchmarks. For larger datasets like MATH500 (500 samples) and AMC (83 samples) in Table 1, the percentages are naturally less "round" (e.g., 80.53%, 56.48%).
> >
> > We hope this clarifies that our reporting is both rigorous and transparent.
> >
> > ---

---

> > > ### Author Response · Authors · 2025-11-20
> > > **Response to Reviewer HA2y - Part 3**
> > >
> > > **Q6.** Regarding the suggestion of including MRR.
> > >
> > > **R6.** We thank the reviewer for this thoughtful suggestion. MRR requires a meaningful ranking of outputs based on real-valued relevance scores. However, in mathematical reasoning benchmarks, each sampled solution is evaluated using a binary exact-match criterion (correct/incorrect). Since all incorrect responses receive the same score and the eight samples are independently generated with no inherent ordering, a valid ranking cannot be constructed. Consequently, standard MRR is not well-defined in this setting and is therefore not adopted in prior work on mathematical reasoning or GRPO-based reinforcement learning (e.g., DAPO, Dr.GRPO).
> > >
> > > ---
> > >
> > > **Q7.** The use of quotation marks.
> > >
> > > **R7.** We sincerely thank the reviewer for the careful and helpful observation. We have thoroughly checked the entire manuscript and corrected all instances of incorrect quotation marks, replacing '' with the proper typesetting ``. These fixes are reflected in the revised version.
> > >
> > > ---
> > >
> > > **Q8.** The placement of the Related Work section.
> > >
> > > **R8.** We thank the reviewer for the constructive feedback and apologize for the placement of the Related Work section. Following the reviewer's advice, we have moved a streamlined and more focused version of the Related Work section into the main paper, ensuring that the key contextualization appears directly in the core content. These changes are reflected in the revised manuscript.
> > >
> > > ---
> > >
> > > **Q9.** Why / how did you choose a value of 16 for the number of samples per prompt?
> > >
> > > **R9.** We selected the group size of $G=16$ to align with the prevalent experimental setups in recent RLVR and GRPO literature. This value strikes a standard balance between estimating a stable group baseline and maintaining computational efficiency. Specifically, $G=16$ is adopted by:
> > >
> > > [1] DAPO (Yu et al., 2025).
> > >
> > > [2] Skywork-OR1 (He et al., 2025).
> > >
> > > [3] FR3E (First Return, Entropy-Eliciting Explore; Zheng et al., 2025).
> > >
> > > By adhering to $G=16$, we ensure that our implementation is consistent with these SOTA baselines, minimizing the impact of hyperparameter variance on the comparison.
> > >
> > > ---
> > >
> > > **Q10.** Are the values for clipping in DAPO the standard ones?
> > >
> > > **R10.** Yes, the clipping thresholds for DAPO (specifically $\epsilon_{low}=0.2$ and $\epsilon_{high}=0.28$) were not tuned by us but were adopted directly from the official DAPO paper and codebase. We used their official recommended configurations to strictly ensure a fair and accurate reproduction of the DAPO baseline.
> > >
> > > ---

---

> ### Comment · Reviewer_HA2y · 2025-11-23
> **Thanking the authors for their rebuttal**
>
> The authors have responded to my questions well and I'm happy to give a stronger score based on their rebuttal.

---

> > ### Author Response · Authors · 2025-11-23
> > **Thank you for your recognition and score update**
> >
> > Dear Reviewer,
> >
> > We sincerely thank you for your positive acknowledgment of our rebuttal and for your willingness to raise the score. We truly appreciate your thoughtful feedback throughout the review process. Your constructive comments helped us further improve the quality of the paper.
> >
> > Sincerely,
> >
> > The Authors

---

### Author Response · Authors · 2025-11-20
**General Response to Area Chair and Reviewers**

We thank the Area Chair and reviewers (HA2y, vQ2H, YdgL) for their efforts and constructive feedback. We are encouraged by the strong consensus regarding the significance and contributions of our work. In particular:

**Insightful Problem Analysis.**
All reviewers affirmed the importance of identifying the *“Singularity Trap.”* Reviewers HA2y and YdgL specifically praised our causal analysis explaining why entropy regularization fails under extremely long-tailed vocabularies and gradient singularities.

**Sound and Novel Framework.**
The proposed Info-GRPO was recognized for both its theoretical grounding and its novelty. Reviewer YdgL highlighted the “new perspective” introduced by our information-theoretic formulation, while reviewer vQ2H emphasized that the method is “simple and intuitive” as well as “easy to reproduce.”

**Robust Empirical Results.**
Reviewers acknowledged the strong empirical evidence. Reviewer vQ2H noted the “consistent improvements” across settings and the “thorough ablation studies,” and reviewer HA2y found our visualizations (e.g., Fig. 4) highly effective in motivating the method.


In the following, we address all questions in detail and provide additional experiments incorporated into the revised manuscript.

---

### Author Response · Authors · 2025-11-30
**Summary Comment to the Area Chair**

We deeply appreciate the efforts of the ICLR Program Chairs and fully respect the decision to revert reviewer scores due to the unexpected leakage incident. We support the conference’s commitment to maintaining fairness and academic integrity. We are grateful that our rebuttal and discussions remain visible to the Area Chair, and we hope the following summary will assist your evaluation.

---

**Reviewer Engagement and Score Updates**

Two reviewers have already responded positively to our rebuttal:

**Reviewer HA2y (Confidence 4)**:
  On 23 Nov 2025, the reviewer stated:
  *“The authors have responded to my questions well and I'm happy to give a stronger score based on their rebuttal.”*
  The reviewer subsequently increased the score **from 4 to 6**.

**Reviewer YdgL (Confidence 3)**:
  On 26 Nov 2025, the reviewer confirmed:
  *“Thanks for the authors' response; my review remains the same (6: marginally above the acceptance threshold).”*
  The reviewer retained a positive score **of 6** and acknowledged the value of our additional experiments.

**Reviewer vQ2H (Confidence 4)**:
  While this reviewer has not yet posted a follow-up response, **their core questions (Q1 and Q5) overlap substantially with Reviewer YdgL’s Q3 and Q1, which have already been addressed to the satisfaction of Reviewer YdgL**. We have also provided detailed replies to all of this reviewer’s remaining points and incorporated these clarifications into the revised manuscript.

These interactions indicate that our responses have effectively addressed reviewer concerns and strengthened their assessment of our submission.

---

### Meta-Review · Area_Chair_3MzN · 2026-01-07

**Summary:**

This paper examines the tradeoff in RL (GRPO) conflicting objectives with respect to diversity/entropy (that favour exploration) versus the collapse in singular high-entropy configurations. The authors then construct a regularisation framework to apply to GRPO that conditions on a set of latent variables to add additional constraints to the cost function. The experiments have been carried out on some math datasets.

**Reviewer Concerns:**

1. The novelty of the work is limited, compared with other works [1-3].
2. Only evaluated on Qwen models.

**Reviewer Scores:**

The paper received diverse ratings initially. The reviewers appreciated that the paper provided an analysis to motivate the method and achieved good empirical results. However, there are main concerns regarding the novelty of the work (especially compared with many other co-current work) and only evaluated on the Qwen model family. After the rebuttal, some of the minor issues have been addressed. The main issues still hold. The final decision is rejection.

---

### Decision · Program_Chairs · 2026-01-26

Reject